# Occupant-centered optimization framework to evaluate and design new dynamic shading typologies

Victor Charpentier[1]*, Forrest Meggers[2,3], Sigrid Adriaenssens[1], Olivier Baverel[4]

1 Department of Civil and Environmental Engineering, Princeton University, Princeton, New Jersey, United States of America, 2 School of Architecture, Princeton University, Princeton, New Jersey, United States of America, 3 Andlinger Center for Energy and the Environment, Princeton University, Princeton, New Jersey, United States of America, 4 Laboratoire Navier, UMR 8205, École des Ponts, IFSTTAR, CNRS, UPE, Champs-sur-Marne, France

* viccharp@gmail.com

**Data Availability Statement:** All relevant data are within the manuscript or its Supporting Information files.

**Funding:** Funded study: VC, FM, SA, National Science Foundation,, Grant No 1538330

**Competing interests:** The authors have declared that no competing interests exist.

## Abstract

Dynamic solar shading has the potential to dramatically reduce the energy consumption in buildings while at the same time improving the thermal and visual comfort of its occupants. Many new typologies of shading systems that have appeared recently, but it is difficult to compare those new systems to existing typologies due to control algorithm being rule-based as opposed to performance driven. Since solar shading is a design problem, there is no single right answer. What is the metric to determine if a system has reached its optimal kinematic design? Shading solutions should come from a thorough iterative and comparative process. This paper provides an original and flexible framework for the design and performance optimization of dynamic shading systems based on interpolation of simulations and global minimization. The methodology departs from existing rule-based strategies and applies to existing and to complex shading systems with multiple degree-of-freedom mobility. The strategy for control is centered on meeting comfort targets for work plane illuminance while minimizing the energy needed to operate space. The energy demand for thermal comfort and work plane daylight quantity (illuminance) are evaluated with Radiance and EnergyPlus based on local weather data. Applied to a case study of three typologies of dynamic shading, the results of the methodology inform the usefulness and quality of each degree-of-freedom of the kinematic systems. The case study exemplifies the iterative benefits of the methodology by providing detailed analytics on the behavior of the shades. Designers of shading systems can use this framework to evaluate their design and compare them to existing shading systems. This allows creativity to be guided so that eventually building occupants benefit from the innovation in the field.

## 1. Introduction

The façade of a building is the interface between the outdoor and indoor spaces. The complexity of designing a façade comes from how different both are and from the strong psychological

influence of the outdoor on the indoor space. This influence requires that the two spaces be mediated by the façade. The outdoor space has extreme variability of environmental conditions. Luminosity and temperatures change daily due to sun movements and to rapidly evolving weather patterns. They also vary seasonally due to the orientation of the earth's axis. In contrast with this extreme variability, the comfort requirements in the indoor space are relatively consistent. Occupants tolerate some variability of thermal conditions based on seasons [1] but overall the neutral thermal sensation broadly remains within a narrow 10°C range of 20°C to 30°C [2]. Visually, there is a strong belief that natural daylight is beneficial for health and productivity [3], but high levels of daylight are also viewed as unpleasant and denote the strong psychological link between daylight levels and visual comfort [3]. In addition, recent results suggest the presence of natural daylight affects the thermal response of building occupants, but only at a psychological level rather than at a physiological one [4]. Visual discomfort in the form of direct sunlight on one's direct perimeter of view is the dominant factor for an individual taking mitigating actions with the façade [3]. The parameters of individual comfort are interdependent due to psychological factors. There is therefore great significance in using a holistic approach to façade design revolving around the improvement of the overall comfort of occupants. The design of façade systems must consider these interactions to increase the perceived value of buildings for occupants. The complexity of both mitigating dynamically changing environmental stimuli (continuous sun movement, evolving cloud coverage) and human comfort requirements (thermal and daylight) can be tackled by external dynamic shading systems. They are façade devices that operate at room level, which is preferred for mitigating thermal and luminous conditions simultaneously [5]. Their operation can be passive (based on a predetermined schedule) or active (based on sensory feedback) and they can include a manual override mode to let the occupants take control of the operation.

Existing strategies for active control of dynamic shades are rule based. They are entirely deterministic methods that always assigns an action to mitigate threshold values of a controlling environmental parameter. Usually, rules are based on measuring the beam illuminance or beam solar radiation on the window [5, 6], or illuminance levels on the work plane [7]. The reference active control strategy is to close the shades when the target environmental parameter exceeds a given upper threshold and to then reopen the shades once the parameter falls below the lower threshold [5, 6]. More recent control algorithms can include rules on the presence of occupants [5] and use of two or more parameters for thresholds [8]. In the case of venetian blinds, the possibility to tune the slat angles with respect to sun angles [8] provides a strategy to eliminate penetration of direct sunlight in the room with the objective to reduce glare. These strategies were implemented early on in [9] for venetian blinds and controlled average illuminance levels on a work plane. These strategies generally treat one parameter as the controlling one and others as indirect consequences (e.g. the energy consumption necessary to maintain the thermal condition of the space). New control methodologies are referenced in two reviews [10, 11]. The authors highlight the improvement of rule-based and sensor-based methodologies to include more complex criteria. Mentions of multi-objective optimization refer not for control of dynamic shades but to design of fixed external shading systems [12–16], positioning of interior movable shading [17, 18] and to the overall design of the façade [19].

The main disadvantage of current control systems is that each typology of shading system has its own rule-based control. Many new systems have appeared recently [20, 21] and those do not yet have precise rules for their control. While control strategies remain mostly rule based, new typologies of external dynamic shades are appearing, as referenced in [11, 22]. Typologies inspired by plants (biomimetics) [23–25] or origami [26] open up the field of what dynamic external solar shading could be. Although they have appeared in the past decade,

their performance has not been modeled systematically as evidenced by the lack of references related to the control of novel external dynamic shading systems in [11]. The assumption guiding the development of more complex shading kinematics is that the added complexity will widen the dynamic shading design field and provide better performances. However, the origins of the kinematic systems and their actual performance are rarely discussed. Existing methods do not allow the precise kinematic control and evaluation of individual degree of freedom. In this context, the innovative designs are often based on the intuition of experienced designers rather than on a data driven design approach. There is currently no systematic method to compare the performance of the most innovative dynamic shading systems. The assumption that the novel dynamic systems improve the performance of facades can only be verified if they can be analyzed and compared to existing solutions.

In this article we depart from rule-based control and we introduce a novel non-deterministic methodology for the comparison and design of dynamic shades based on the global optimization for best performance. To do so, we present a two-step methodology based on the interpolation of pre-calculated energy and daylight simulations and on the search for global energy minimum with strict conditions of occupant comfort. This methodology represents a departure from rule-based control of shades because it allows to easily include shading systems with more complex geometries and makes simulated results the foundation for finding the best position of the shades. In opposition to rules that must be adapted to each shading geometry, our methodology can be applied to any geometry that can be modeled. This article specifically focuses on how to integrate optimization techniques in the design and development of dynamic shades, as opposed to be a simulation-centric article. Therefore, this article explores the first the methodology and then the the results of the presented case study Following a precalculation process [27], the heating, cooling and lighting energy and the work plane illuminance are calculated for all possible shading positions and all time steps. These discrete simulations are then linearly interpolated to create a continuous map between the state of actuation and the performance metric. By doing so, the simulated performances can now be integrated into an optimization routine by genetic algorithm that finds the best position of the shades by searching for the global minimum of energy demand with user-inputted requirements on average and maximum work plane illuminances. In addition, since psychological aspects are key to individual comfort, the proposed methodology can be improved to integrate new metrics of performance.

The case study simulations are based on the actual weather file of Mercer County, New Jersey, USA, to show that this optimization methodology can yield optimal positions in a wide variety of weather conditions such as overcast days with dominating diffuse light. Three types of external dynamic shades are simulated: awning, horizontal slats (referred to as venetian shades) and spherical tracking shades. The awning and the venetian dynamic shades have been selected due to being standard typologies of shades. They support the applicability of our methodology for the comparison and design of existing and novel geometries such as a spherical tracking shading system.

The main objectives of this article are to introduce the methodology and to document how a more advanced type of design process can be used to select or improve dynamic shading devices. This article presents representative results about shading control and analysis. The case of a fictitious building room is selected in Mercer County, New Jersey, USA for three orientations (east, south east and south). Energy demand for heating, cooling and electric lighting are calculated to maintain a fixed level of occupant comfort (thermal and lighting). Work plane illuminance (average and maximum) is calculated as principal metric for daylight quantity. The impact of three different shading systems is shown using the presented methodology and the significance of the results as well as future implementation are discussed. As additional

comfort requirements are included in the optimization routine, this article shows that there is potential to augment what can be done by dynamic shading. Eventually, this could lead to the design of shading systems that provide more value for occupants.

## 2. Methodology to evaluate and compare dynamic shading typologies

The presented methodology provides precise details for the performance evaluation of dynamic shading systems designs. Diverse typologies of shading systems can be evaluated by the same benchmark. The classical approach to shading control is the use of energy thresholds, daylighting thresholds or geometric rules based on shading surface projections. The evaluation algorithm presented in the methodology is not based on rules but on actual measured performance. Shading systems of 1, 2, 3 or more degrees of freedom can be simulated and compared seamlessly with the pseudo-static model of the dynamic behaviour (see Section 2.3). The output of the methodology is three-fold for each time step: position of the shade (decomposed into each degree-of-freedom if the system has more than one), predicted energy consumption of the building system to respect thermal comfort constraints and prediction of whether the visual comfort criteria are satisfied. From this hourly data, several performance indicators can be derived. The main indicators used in this study to quantify the performance of the shades are the total and detailed energy consumption, the frequency that the visual comfort constraints are respected and the standard deviation of each degree-of-freedom's actuation. The latter criterion is defined for a shading system with N degree-of-freedom as a N-vector of the standard deviation of each actuator's position (between 0 and 1) over the analysis period. This indicator provides a powerful design feedback, if the standard deviation of one of the actuators is low, the degree-of-freedom of the shading system is not useful. The shading system can therefore be simplified by removing the actuator.

### 2.1. Schematic presentation of the methodology

There are five steps to the methodology for the assessment of the performance of shading devices. The success of the process is conditioned on the prior definition of the objectives (e.g., minimization of energy demand) and constraints (e.g., visual comfort, thermal comfort) of the analysis. The final step of the methodology is the decision-making stage. In this step, the choice of whether the results produced by the analysis are satisfactory or not, should be made. As presented in the case study, the design of a shading system is better performed by comparing the results of several systems with one another. While a type of shading might satisfy the constraint of daylighting, it is possible that its design be refined to reduce the energy consumption. In this case, the analysis and optimization should be run again with a modified version of the shading system to test. This iterative design can lead to higher performance shading systems. There is no end to improving the design of a shading system so the methodology can theoretically be looped forever. The design of a shading system should be stopped when the result is deemed good enough, per the requirements of the project or by the designers understanding of the situation.

The methodology is sequenced as follows (see also Fig 1).

1. Design of the shading system / Selection of comfort constraints values

2. Core analysis

    a. Evaluate the metrics selected for the analysis at each actuation step, at each sun hour

    b. Model the behavior of the shade with interpolation

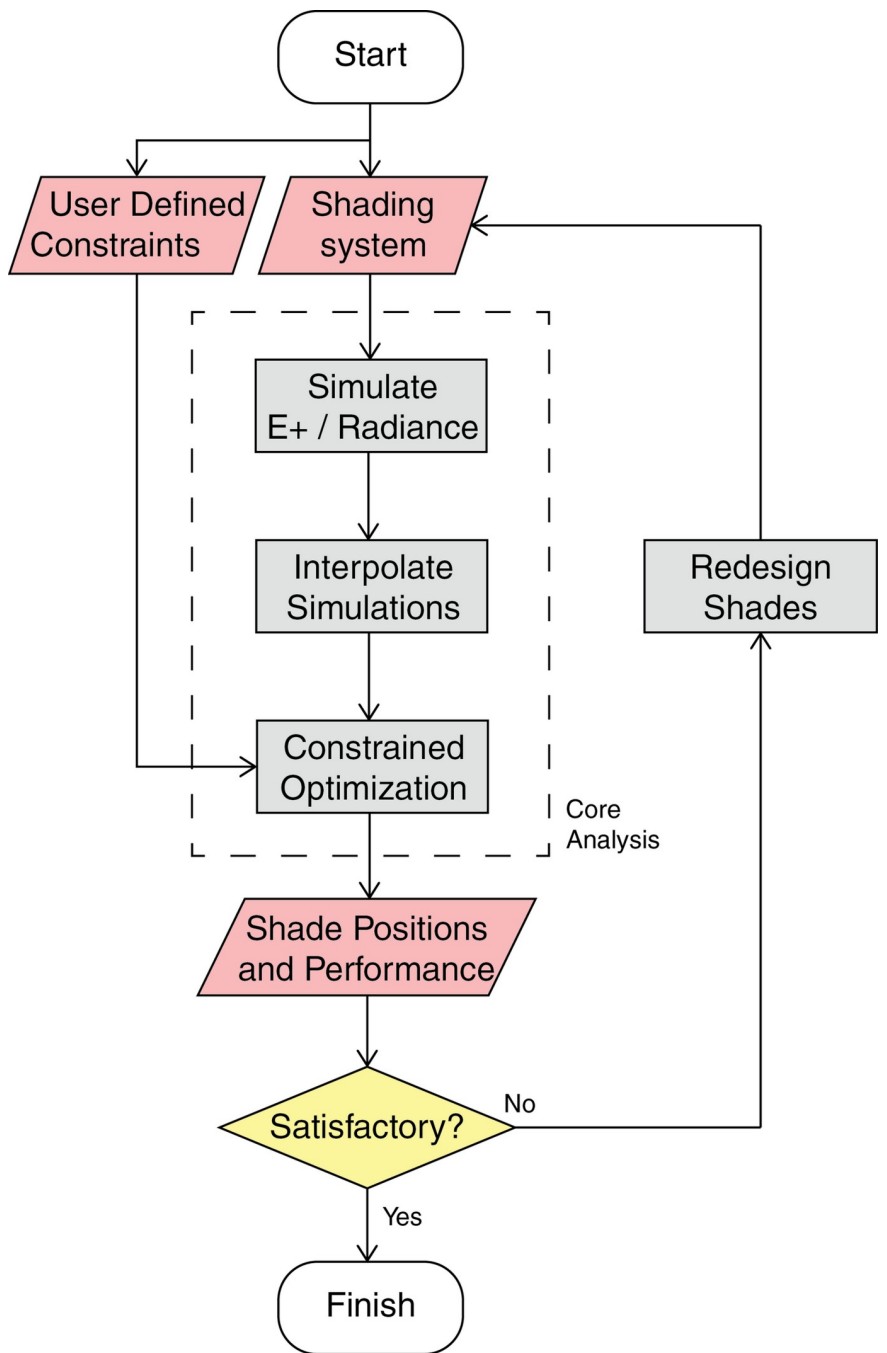

**Fig 1. Flowchart of the methodology for analysis and iterative design of shading systems.** Grey boxes denote action to be taken; red boxes denote states of the system and the yellow box tests the performance of the shades to the objectives of the designer.

 c. Optimize shade position with comfort requirements

3. Results satisfy the constraints of comfort and performance.

 a. Yes: Analysis is over

b. No: Loop back to step 1 for re-design.

## 2.2. Selected energy and daylight control and comfort variables

Five variables are selected to characterize the performance and comfort of the shades: the heating energy demand, cooling energy demand, electric lighting energy demand, average and maximum illuminance on an 80-cm high work plane. The energy and daylighting simulations are performed with DIVA [28] in the program Rhino3D/Grasshopper [29]. The thermal state of the test room is simulated with EnergyPlus [30]. The daylighting and electric lighting analysis are performed by Radiance [31].

**2.2.1. Thermal model.** The goal of the thermal model is to produce energy demands for heating and cooling of a building space. The EnergyPlus simulation engine is interfaced by Archsim in Grasshopper3D/Rhino3D. The model takes the climate data from an epw file as input. The thermal comfort of the space is dictated by the air temperature only. The heating setpoint is set at 20˚C while the cooling setpoint is set at 26˚C. The heating set-back is set at 15˚C while the cooling set-back is set at 32˚C.The heat balance is performed with four time-steps per hour and implements the conduction transfer function method. The calculation of solar radiation is performed with the detailed timestep integration method. In this study, the thermal model in EnergyPlus is not set to integrate electric lighting and daylighting simulations. Both of those are run independently in Radiance. EnergyPlus uses the Sutherland Hodgman polygon clipping algorithm [32] to determine the projection of the shading modules on the window. This algorithm does not support concave polygon shadows [33].A shading system of complex geometry could be form a concave polygon once it is projected. The impact of this algorithmic limitation is not studied here but should be investigated further.

**2.2.2. Daylighting model.** The illuminance on the work plane is used as a metric for assessment of daylight quantity. Two criteria are derived from the raw illuminance on the work plane to quantify the daylight quantity: the *average illuminance* ($E_{average}$) and the *maximum illuminance* ($E_{max}$). The visual comfort parameter are given by values from literature for both parameters. An average illuminance of 500 lx is recommended for the work plane for paperwork or computer work [9, 22, 34, 35]. This value can also be found to be between 300 and 500 lx in other references [36, 37]. In the optimization procedure, the target illuminance for average illuminance is set at 500 lx. A maximum illuminance over 2000 lx is likely to cause user visual or thermal discomfort [38–40]. Some studies on the matter of useful daylight illuminance (UDI) correlate the occurrence of glare to values of illuminance over 3000 lx [41]. In this study, however, the maximum value of illuminance of 2000 lx is considered as the upper limit of daylight comfort.

**2.2.3. Electric lighting model.** The average illuminance provided by daylight is not uniform over the work plane and will sometimes not be enough to provide comfortable ambient light conditions. Therefore, electric lighting is modeled to predict the demand of electric energy required to complement the optimized daylight provided by each shading system. A typical lighting system with 12 W/m$^2$ of power is assumed for the simulation. The control system considered is a dimmer with occupancy on/off control. The setpoint of the lighting system is 500 lx. The control algorithm dims the lights proportionally to the difference between the actual average of the sensors and the objective. When the daylight is 500 lx over the sensor, the dimmer is set to its lowest value (10% of power). The lighting at a given moment is determined to provide enough light to reach the setpoint of the system uniformly over the work plane. The default Archsim electric lighting is used. Given the geometry of the room and the presence of a large window on one wall only, daylight diminishes with the depth of the room.

The algorithm for electric lighting is open loop; it does not feed back into the daylighting assessment. The electric lighting energy was calculated with the goal to meet the average

lighting constraint on the work-plane. If the average work plane illuminance provided by natural daylight is below the 500 lux required by the constraint, the electric lighting compensates with the appropriate amount of lighting to reach the target 500 lux. The electric lighting is provided uniformly to the work plane so the average illuminance will increase to 500 lux. It is not a multi-zone lighting and cannot compensate for the diminution of daylight with the depth of the room. That more advanced feature should be integrated in further lighting specific studies. Since the simulation is open loop, the algorithm does not iterate to evaluate the final work plane illuminance. In that sense, it is similar to the open loop operation system described in [11]. The final lighting distribution would be found by iterating over the sum of the daylight and electric lighting contributions. The current algorithm guarantees that the average illuminance on the work plane of this new lighting (electric and daylight) is 500 lux.

## 2.3. Pseudo-static approximation of the dynamic behaviour of the shades

The precise positioning of the shades at each time step for an entire year is the basis of the presented methodology. For the optimization algorithm to determine these positions it must search all the combinations of successive positions to find the one combination that satisfies the constraints and minimizes the energy demand. A full model of the system requires that for a given time step, each of the possible positions for the following time step are simulated. This requirement is fixed by the physicality of the model. Thermal energy is stored in materials. There is therefore thermal lag between each time step. Controlling the shades dynamically takes a high amount of computation resources since it means accounting for the thermal lag that is carried from one step to the next for all the possible shading positions. For a given step k, the thermal load at the end of the step should be carried to step k+1 that can have N different shading positions. There are 4306 hourly steps in the weather file used for the case study. The total number of combinations c that would need to be simulated is c = 4306 N. For instance, the two d.o.f. system used in the case study has 81 different positions. The total number of thermal simulations to run in this case would be c = 348 786.

To reduce the simulation cost, the dynamic behaviour of the shading systems is simplified. For each shading system, the range of actuation is decomposed in discrete positions. Each position is treated as a fixed shading system. A shading system with one degree-of-freedom decomposed in six actuation positions will be simulated as six independent fixed shading systems. The two degree-of-freedom system mentioned in this paragraph is simulated as 81 independent fixed shading systems (9 positions for each d.o.f.). Each actuation position of the shading systems is evaluated for the time period similarly to a fixed shading system.

The dynamic behavior is recreated artificially by an interpolation step. Specifically, for each time step, the results of the simulation for the equivalent fixed shading systems are linearly interpolated. An analytical function is produced at each time step for each of the five simulated variables. This interpolation creates a pseudo-dynamic behavior that reduces the total number of simulations. The two d.o.f. system previously mentioned is simulated with 81 calculations (as opposed to 348 786).

Thermal lag is included in the calculation since a full thermal model is run. But there is an imprecision on the magnitude of this lag. This imprecision is bound to exist due to the simplification of the calculation. There is no lag in terms of daylight, so this issue only concerns the thermal behavior of the space.

## 2.4. Interpolation of simulated results for behavior modeling

This study uses an optimization algorithm to determine the behavior of a shading system. The algorithm searches the best position of the shades at a given time step considering an objective

and some constraints. The algorithm searches the optimal position in the domain of actuation from 0 to 1. This domain is continuous, i.e. any and every position of the shades in the domain can be realized (see Table 1 for each shading system domain of actuation). However, the result of simulations for adaptive shades is discrete. For a given sun hour and building orientation it is only possible to evaluate the effect of a dynamic shade one position at a time. There is therefore a discrepancy between the continuous functions needed by the optimization algorithm and the discrete results of simulations.

To bridge that gap, this study introduces the use of interpolation. From the simulations' results of a small number of positions sampled in the domain, the continuous behavior of the shades is reconstructed. This pseudo-dynamic model can be applied to any dynamic shading system. In the case study, awning and venetian shades are simulated at nine positions of the actuator. The spherical tracking shades are simulated at 81 positions (nine per d.o.f.). The heating, cooling and lighting energies are interpolated as well as the average and maximum illuminances on the work plane (Fig 2). In Fig 2, the simulation results are shown as red squares and the interpolated functions are the black lines (one d.o.f.) or surfaces (two d.o.f.). The interpolation is performed in MATLAB using a linear interpolant object (griddedInterpolant function). The interpolant is an interpolating function that can be evaluated at query points. It is easily integrated into analytical optimization systems. The actuation domain is sampled at regular intervals over the actuator's range of motion, which creates a uniform grid for interpolation (see the regular sampling on Fig 2). The interpolation is chosen to be linear. It provides a $C°$ continuity to the function, which is satisfactory for this optimization. The linear interpolation function is less smooth than a higher degree polynomial or spline interpolation, but it was selected because it has the benefit of limiting errors when large values of illuminance occur. For instance, if the simulation produces an extremely high result at a point of the actuation grid, the linear interpolation would not propagate the error to neighboring points. The values of illuminance can peak sharply and locally to high orders of magnitude (e.g., from $10^2$ lx to $10^4$ lx), it is therefore beneficial to constrain the response locally, hence linear interpolation is preferred.

The number of interpolated functions stored is independent of the number of d.o.f. of the shading system. For a given sun hour, the five parameters (heating, cooling and lighting energies and average and maximum illuminance) are interpolated. There is therefore 21 530 (= 5 parameters*4306 sun hours) functions stored for each shading system.

## 2.5. Optimization system for control of shades

The goal of the optimization system is to minimize energy demand (heating, cooling, and lighting) under constraints thermal and visual comfort. In section 2.2, the five parameters of the optimization are described with analytical relationships based on the simulated results: heating energy, cooling energy, lighting energy, average work plane illuminance and

**Table 1. Mobility and range of motion of the shading systems.**

| | Shading system | Degree-of-freedom ID | Description | Range | | Step Count | Step Size | No. possible positions |
|---|---|---|---|---|---|---|---|---|
| | | | | Min. | Max. | | | |
| 0 | Unshaded | 0 | Unshaded glazing | - | - | - | - | 1 |
| 1 | Awning | 1 | Roller extension | 0 | 1 | 9 | 0.11 | 9 |
| 2 | Venetian shades | 1 | Angle of individual slats | 5° | 70° | 9 | 7.22° | 9 |
| 3 | Spherical tracking shades | 1 | Elevation angle | 0° | 70° | 9 | 7.78° | 81 |
| | | 2 | Azimuth angle | -45° | 45° | 9 | 10° | |

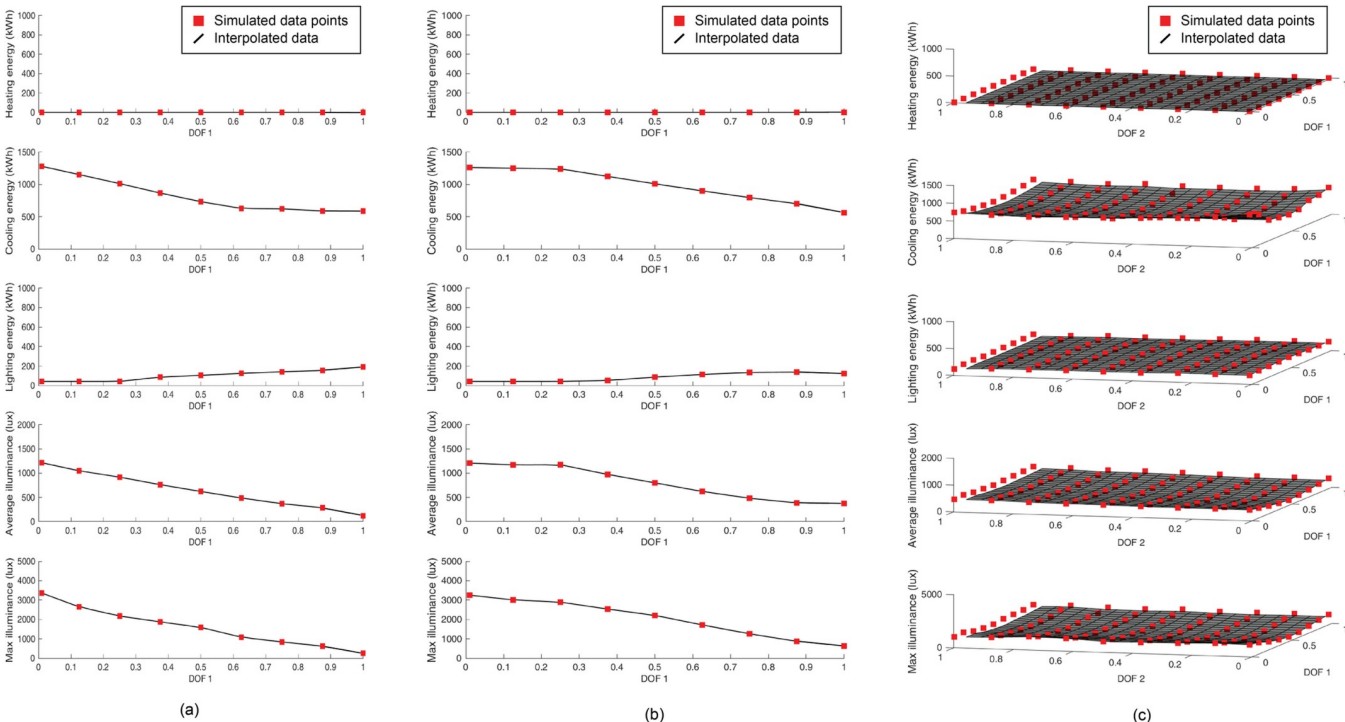

**Fig 2.** Linear interpolation on heating, cooling and lighting energy and on average and maximum illuminance for 1D actuated awning shades (a) and venetian shades (b) and for 2D actuated spherical tracker shades (c) data for July 6 at 12h00 –East orientation.

maximum work plane illuminance. Thermal comfort is considered by the setpoints of the thermal analysis performed in EnergyPlus. The heating temperature setpoint is set at 20˚C, which means the temperature of the room will not go below 20˚C when the weather is cold. The cooling temperature setpoint is set at 26˚C which means the temperature will not go above 26˚C when the weather is warm. Those two temperature setpoints guarantee the thermal comfort of the occupants. No matter what the shading typology is, no matter if it's the baseline case, the thermal comfort is maintained by the setpoints and a room temperature in that range is expected at any given moment. The shading systems will have an impact on the amount of energy necessary to maintain this setpoint temperature. Therefore, thermal comfort is implicitly guaranteed in this methodology.

In opposition, the visual comfort is an explicit constraint of the optimization. The daylight level is calculated explicitly from the positions of the shades. It is not a product of the Energy-Plus simulation. The visual comfort is coupled with the thermal analysis since each shading systems will have several possible actuation positions and each position influences both the thermal load and the level of daylight transmitted to the room. The assumption with the thermal comfort is that the setpoints temperatures are by default respected. Therefore, thermal comfort is not the driver of the optimization. In other words, for any solar radiation or shading situation, with enough energy inputted in the system, air temperatures that respect the setpoints can be achieved. However, not all these situations will lead to satisfying visual comfort. Therefore, visual comfort is the controlling constraint in the optimization.

The interpolated functions are used to solve the constrained optimization system ($S$). The global minimum method called Augmented Lagrangian Genetic Algorithm (ALGA) in Matlab (ga function) is used. The formulation of the optimization system ($S$) in Eq 1 refers to a single

solar position s $s$ and the shading system $\alpha$.

$$(S)\begin{cases} \min_{x\in I}(E_{heat.s}^{\alpha}(x) + E_{cool.s}^{\alpha}(x) + E_{light.s}^{\alpha}(x)) \\ \\ s.t.\begin{cases} A_s^{\alpha}(x) = l_1 \\ \\ M_s^{\alpha}(x) < l_2^+ \end{cases} \end{cases} \qquad (1)$$

with $\alpha$ as the shading system, s the sun vector, I $I$ the actuation interval, $E_{heat.s}^{\alpha}$ the interpolated heating energy for sun vector $s$ s and shade $\alpha$, $E_{cool.s}^{\alpha}$ the interpolated cooling energy for sun vector $s$ and shade $\alpha$, $E_{light.s}^{\alpha}$ the interpolated lighting energy for sun vector $s$ and shade $\alpha$, $A_s^{\alpha}$ the average illuminance interpolated function, $M_s^{\alpha}$ the maximum illuminance interpolated function, $l_1$ the target value for the average work plane illuminance (500 lx) and $l_2^+$ the upper limit for the maximum work plane illuminance (2000 lx). **x** is a vector of dimension the number of d.o.f..

The system is solved for both single and two d.o.f. shades the same way. The difference between the two cases is the actuation interval $I$. In the single d.o.f. case (1D case), the interval $I$ is a segment $I = [0,1]$ while in the two d.o.f. case (2D case), $I$ is a plane $I = [0,1] \times [0,1]$. The constraints tolerance is set to 25 lx to speed up the convergence of the optimization system.

The optimization methodology is adapted to both hot and cold periods since the objective function is the sum of all the energy demand in the system at a point $t$ in time. Similarly, the constraints of comfort are the same for users throughout the year. The optimization system must be solved for each shading system, orientation, and sun hour considered. In total in the case study, the optimization system is solved 38 754 (= 3 orientations*3 shading systems*4306 sun hours) times.

## 3. Detail of the case study

Three types of shading systems are evaluated and compared to a baseline, non-shaded scenario for three façade orientations. Previous studies [42] have demonstrated the usefulness of design choices with thermal and lighting objectives. A study of model-based control of shading [43] applied daylighting objectives to position roller shades adequately. The present study focuses on mediating the energy (heating, cooling, and lighting) demand under lighting requirements for three types of shading systems. Weather data relative to Mercer County NJ (USA, latitude 40.3573˚ N) comes from the typical meteorological year (TMY3) dataset [44]. In the dataset 4306 hours of sunlight are recorded. The outdoor daylight and sky conditions at the location of the study can be seen on Fig 3. The global horizontal illuminance can reach levels up to 100 000 lux during the year, which is much higher than the 2000 lux defined as a threshold for visual comfort.

### 3.1. Façade orientation and room geometry

East, south-east, and south orientations are considered. By symmetry, the western orientation is expected to behave similarly to the eastern orientation in terms of daylight levels. One of the main differences, however, is that thermal mass carryover tends to make afternoons hotter on the west than the east where the night was cool before sun exposure.

The simulated perimeter office is 5 m deep, 4.5 m wide, and 3.2 m high (Fig 4) with a 65% window-to-wall ratio. These dimensions are similar to those found in previous daylighting studies [45, 46]. The window is 2.2 m high ($h_w$) and 4 m wide ($w_w$); it sits at 0.5 m from the ground (Fig 3). The shading system covers the entire window. The work plane is defined by a plane 0.80 m above the interior ground and offset by 0.50 m from the window (Fig 4). The

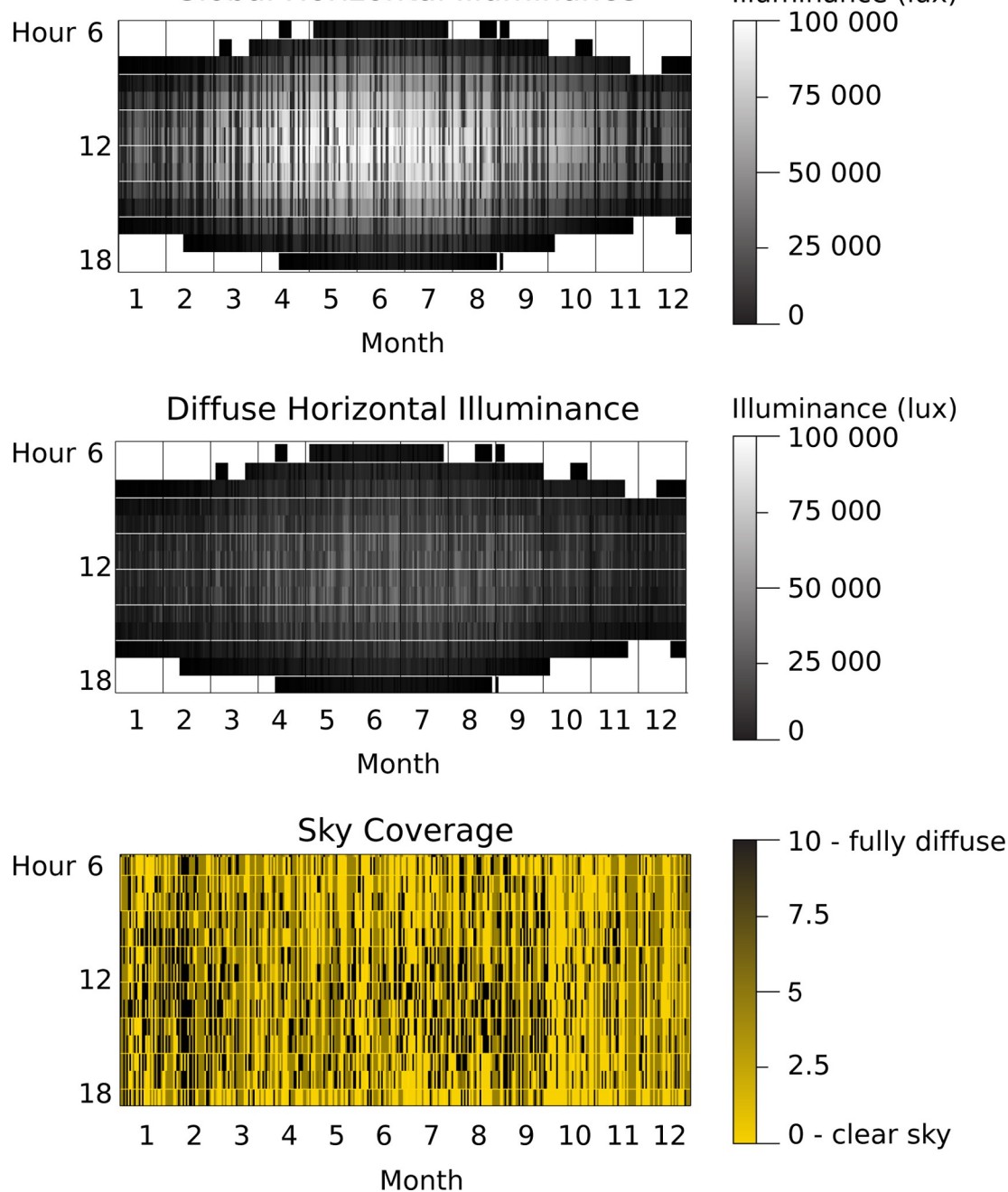

**Fig 3. Outdoor daylight quantity conditions (global and diffuse horizontal illuminance) and sky coverage in Mercer County, New Jersey, USA (10 means the sky is fully diffuse, 0 that the sky is clear).**

choice of dimensions for the work plane will have an influence on the outcome of the study. The work plane chosen starts at 0.5 m from the window, which will lower the average illuminance factor in the analysis. In our case, this work plane was designed in agreement with precedents in the literature [5]. An outside ground plane (30 x 24m) is added to the model (Fig 4) to add ground reflections in the daylighting study.

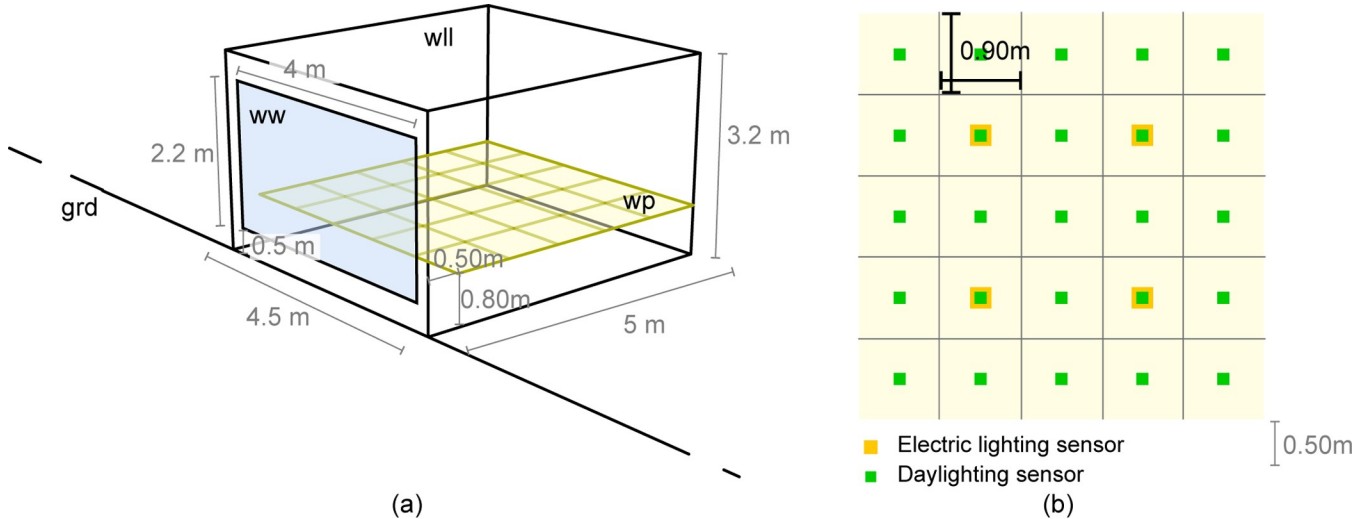

**Fig 4.** (a) Perimeter office space–window (ww) in blue, work plane (wp) in yellow—exterior / interior ground (grd), walls (wll) and ceiling indicated–(b) The grid represents the daylighting analysis grid. There is one daylight sensor per square (green). The four yellow sensors are specific to artificial lighting and are used to determine when electric lighting is needed.

### 3.2. Choice of external shading system: Awning, venetian shades and spherical solar tracking

Three categories of external shades are implemented: typical awnings, venetian blinds, and spherical solar tracking shades. The unshaded window is evaluated in the analysis for a baseline comparison. The awning and the venetian dynamic shades have been selected due to being standard typologies of shades. They show that our methodology can be applied for the evaluation of existing geometries as well as more novel geometries such as a spherical tracking shading system. The spherical tracking shades are controlled using 2 actuators. Therefore, this system has two degrees of freedom. Awnings and external venetian blinds are controlled by a single actuator; hence they are one degree-of-freedom systems (see Fig 5). They have been modeled to resemble commercially available systems [47–50]. Since they are all dynamic, the three shading systems can be described as tracking the sun movements to some extent. However, the typical awning and the venetian shades do not track azimuthal movements of the sun; this reduces the dimensionality of the tracking and hypothetically, limits its performance. Table 1 summarizes the mobility and range of motion of each shading system.

Spherical movements derive from spherical coordinates in which the position of a point in space is described not by $(x,y,z)$ but by $(r, \phi, \theta)$, with $r$ as the radial distance from the origin, $\phi$ the azimuthal angle, and $\theta$ the zenith angle. The third shading system evaluated is capable of spherical tracking motion. It follows both the elevation and the azimuth angle of the sun so that the shade surface can always be oriented perpendicularly to the sun vector, if needed. Such a spherical system has been proposed for photovoltaic collection on façades [51]. The geometry and range of motion of the spherical tracking shades stems from ongoing work of the authors to design a spherical tracker.

Typical awning shades are controlled by the rotation of the roller carrying the fabric. The range of motion of the roller allows the shade to cover the full height of the window $h_w$ (d.o.f. 1). The shade moves with respect to the façade in a circular arc motion as the roller is rotated (see Fig 5A). The 8 slats of the venetian blinds individually rotate (d.o.f. 1) between 5˚ and 70˚ from the vertical window plane. Each slat is 27.5 cm deep such that they cover the whole

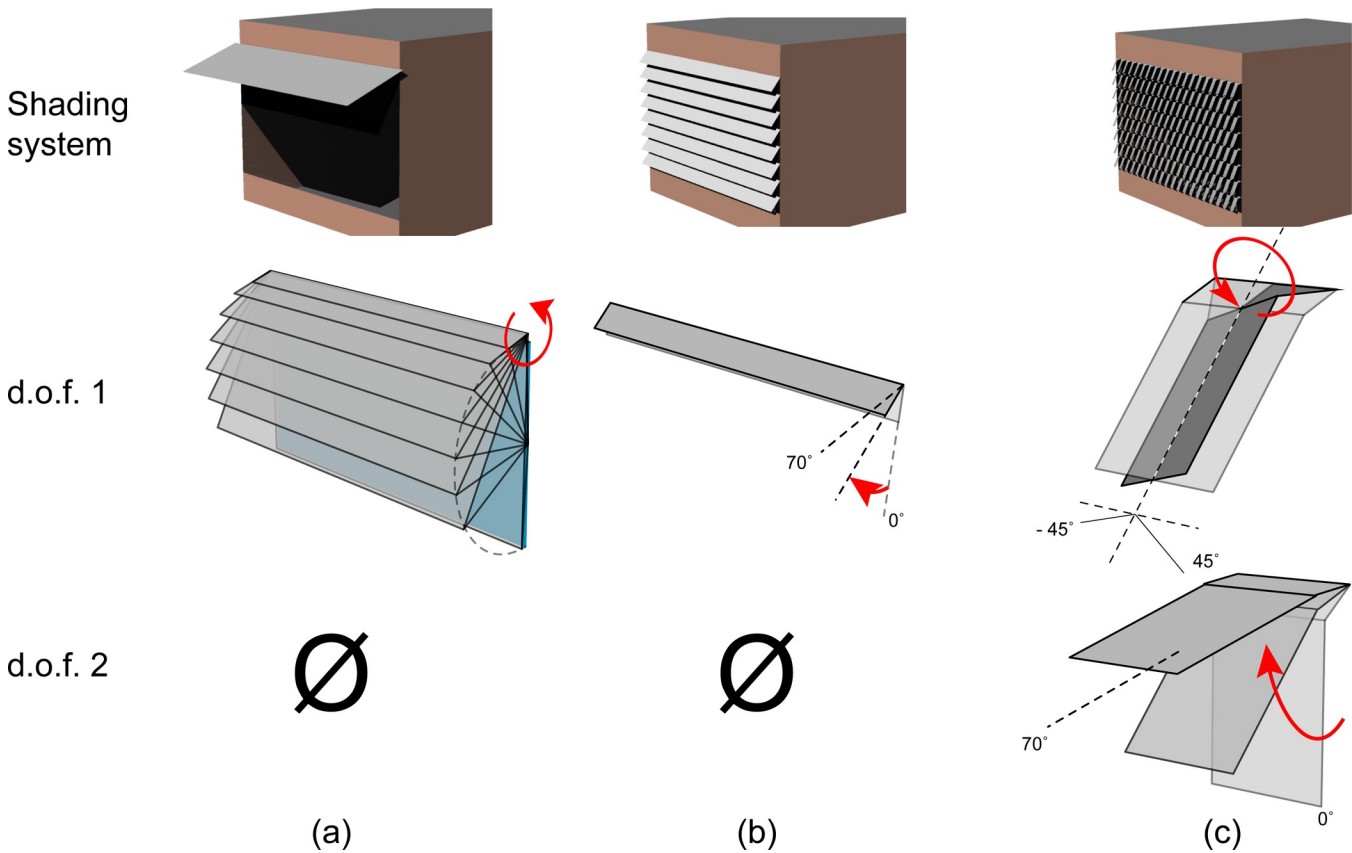

**Fig 5.** Three types of shades and their associated d.o.f. (a) Awning with roller extension (d.o.f. 1), (b) Venetian shades with slat angle (d.o.f. 1), and (c) Solar tracker with azimuth angle (d.o.f. 1) and elevation angle (d.o.f. 2)–circular arrows denote rotational actuators.

window when closed (Fig 5B). Finally, the spherical solar tracking shade rotates with the façade to follow sun elevation in the range [0˚, 70˚] (d.o.f. 2) and rotate longitudinally to track the sun azimuth (d.o.f. 1) with the range [−45˚, 45˚] (Fig 5C). Each element of the tracking system is 20 cm wide and 27.5 cm long. The awning is the only system that is able to provide a completely unobstructed view to the outside. The venetian shades and the spherical tracker system remain in front of the glazing even in the fully open position.

Two d.o.f. increase the range of motion of the spherical tracker but add mechanical complexity. The spherical solar-tracking shade is two d.o.f. by default and is inspired by previous research on plant solar tracking movements [52]. The model of the shade is constructed in two parts (Fig 5). The smaller active part generates the movement and the larger passive part produces the shade. The passive part is a flat surface (Fig 5C). In the case of perfect solar tracking, the passive surface would always remain perpendicular to the sun vector. That is not necessarily the case in our methodology since the positions are derived from energy and daylight considerations.

### 3.3. Detail of the energy and daylighting controls for the case study

**3.3.1. Thermal energy.**    The epw file of Mercer County, NJ is selected as weather input. In this model, the heating and cooling are running every day of the week during the hours of occupancy of the space (Fig 6). Hourly internal gains from people are modeled based on the office occupancy schedule in Fig 6 and vary depending on the hour of the day.

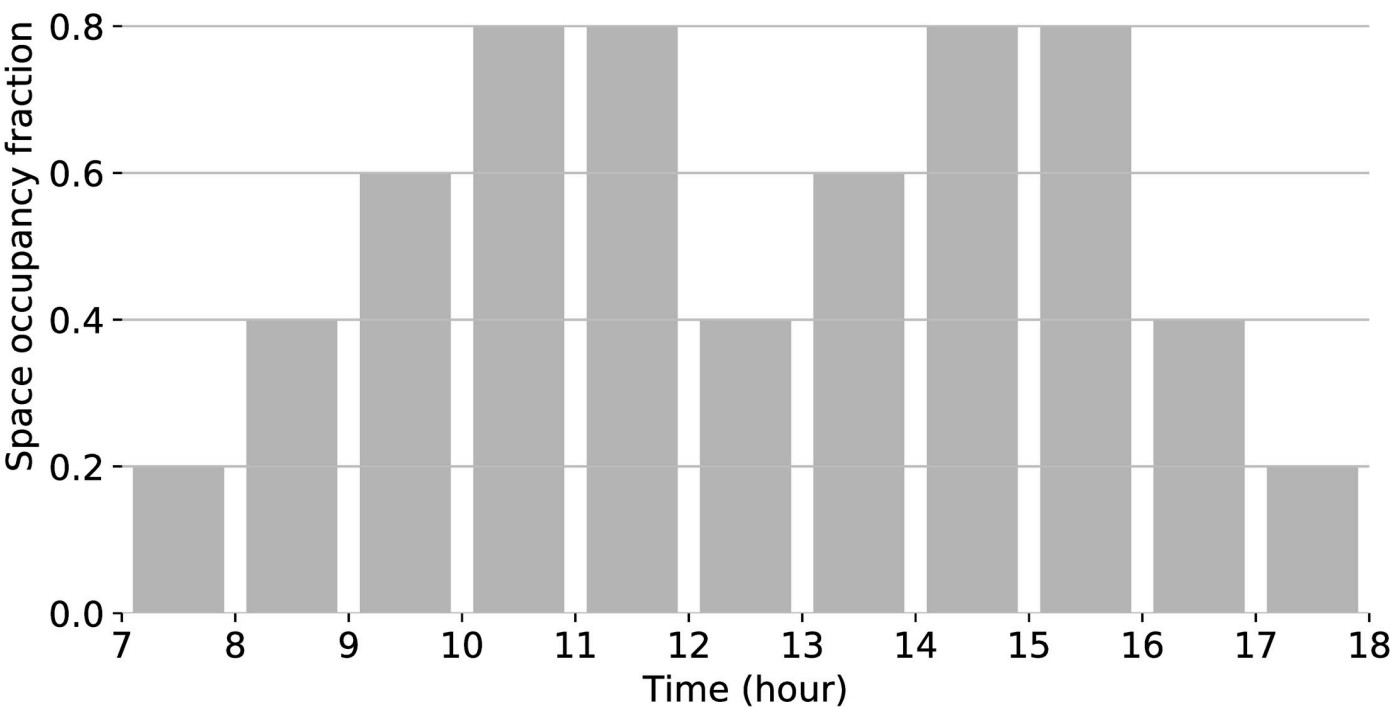

**Fig 6. Occupancy schedule for modeling the presence of people in the test room.** This schedule is only valid for weekdays. No occupants are present during the weekend.

The construction material of the space is detailed as follows: the ground (interior and exterior) is an adiabatic 200 mm thick concrete slab and the walls and roof are a layering of 120 mm thick insulation and 200 mm thick structural concrete. The total thermal resistance (R-value) for the walls and roof is 3.63 K·m²/W. The window is a clear double-paned window with a thermal resistance (R-value) of 13.3 K·m²/W. This value is higher than industry averages (~1 K·m²/W). This high value is the default value in DIVA. The full characteristics of the window are presented in Table 2. For interior convection the TARP algorithm is used. For exterior convection the DOE-2 model is used. Finally, an outside air infiltration of 0.2 air changes per hour (ACH) is implemented.

**Table 2. Characteristics of room materials for Diva daylighting and Archsim/EnergyPlus.**

| | Surface | Daylighting | | Thermal | |
|---|---|---|---|---|---|
| | | **Material Name in Diva** | **Material properties** | **Material Name in Archsim** | **Material Properties** |
| 1 | Wall | GenericInteriorWall_50 | Diffuse reflector with 50% reflexivity | 120mmInsulation_200mmConcrete | R-value 3.63 K.m²/W |
| 2 | Ceiling | GenericCeiling_80 | Diffuse reflector with 80% reflexivity | 120mmInsulation_200mmConcrete | R-value 3.63 K.m²/W |
| 3 | Window | Glazing_DoublePane_Clear_80 | Visual transmittance 0.80 Visual transmissivity 0.87 SHGC 0.764 | DoublePaneClr | R-value 13.3 K.m²/W * |
| 4 | Floor | GenericFloor_20 | Diffuse reflector with 20% reflexivity | 200mmConcrete | Adiabatic |

*Note that the R value for the window is higher than industry standards and corresponds to the default value in DIVA.

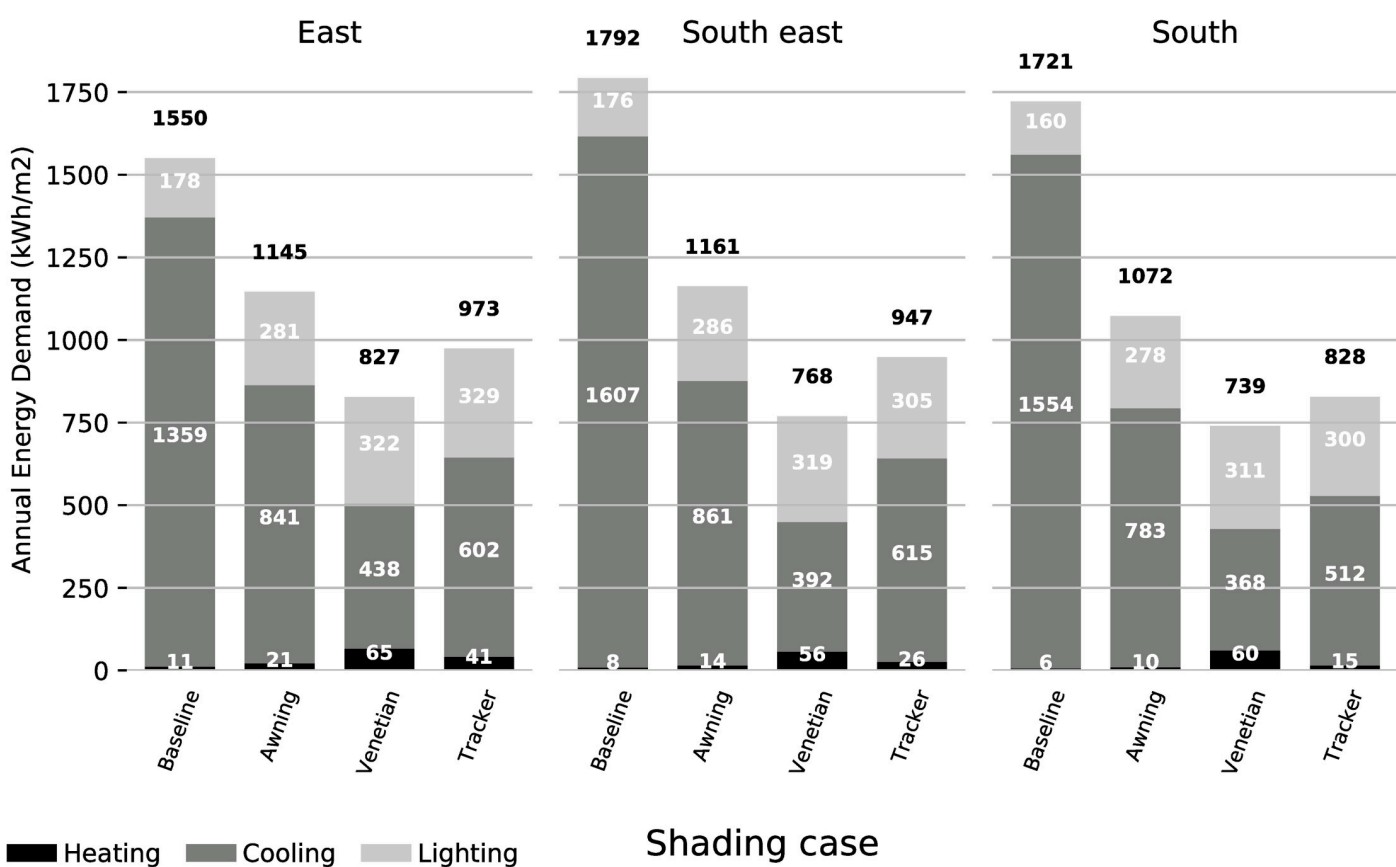

**Fig 7. Annual energy demand in kWh for east, south-east and south orientation and for the baseline case, awning, venetian, and spherical tracker shades.**
Heating, cooling, and lighting energies are to make the total energy demand.

**3.3.2. Daylighting model.**    The illuminance on the work plane is calculated using *Radiance* with 25 grid sensors (one per square in Fig 3B). They divide the work plane (20.25 m$^2$) into 0.9 m x 0.86 m squares. The shades present a 4% transmittance as implemented in [53]. The inside and outside ground have 20% reflectivity. The walls have a 50% reflectivity, while the ceiling has an 80% reflectivity (see Table 2). These parameters are selected to be generic and should be adapted to case-specific studies. The window is a standard double glazing with a solar heat gain coefficient (SHGC) of 0.764 and a visible transmittance of 0.800 (see Table 2). The calculations were performed for Mercer County, USA (latitude 40.3573˚N) for the entire year. The calculation in Radiance has been shown to overpredict the illuminance computed [35], so the results should ideally be verified against experimental data for validation.

**Table 3. Variations of total annual energy demand of the three shading systems to the baseline case.** The results are produced by the optimization process.

| Orientation | Total annual energy demand variation | | |
|---|---|---|---|
| | **Awning** | **Venetian** | **Tracker** |
| East | -26% | -47% | -37% |
| South-east | -35% | -57% | -47% |
| South | -37% | -57% | -52% |

**Table 4. Variations of the annual cooling energy demand of the three shading systems to the baseline case.**

| | Annual cooling energy demand variation | | |
|---|---|---|---|
| Orientation | Awning | Venetian | Tracker |
| East | -38% | -68% | -56% |
| South-east | -46% | -76% | -62% |
| South | -50% | -76% | -67% |

**3.3.3. Electric lighting energy.** As described previously, an objective of 500 lx average illuminance constrains the system. The daylighting grid divides the work plane in 0.9 m x 0.86 m grid. Four sensors are placed in two rows (see Fig 4B) for the electric lighting system. The number of sensors represents a typical number of people in this office space. The occupancy schedule is set to the same hours as the thermal schedule (Fig 6), with the exception that it considers daylight-savings time.

## 4. Results

The results of the simulations are presented below in three sections: the energy demand (Section4.1),the daylight condition of the work-plane (Section4.2) and the quantification of the use of each d.o.f. (Section 4.3). The results are produced by the optimization of the energy demand with constraints of daylight quantity presented in Section 2. The focus of this section is both on presenting the results of the case study as well as on exploring the methodology. The assumptions of the case study should therefore be understood in the context of an investigation of the novel method presented for external dynamic shades.

### 4.1. Mitigation of energy demand for the three types of shades

The overall combined impact of the optimization methodology on heating, cooling, and lighting annual energy demand is reported in Fig 7. For the three orientations observed, the annual energy demand for the baseline case of no shading is significantly decreased by the three shading systems (Table 3). The optimization results show a decrease of annual cooling demand on the east of 26% for the awning, 47% for the venetian and 37% for the spherical tracking shades. This decrease is more pronounced on the south-east and south: on the south-east—35% for the awning, - 57% for the venetian, and—47% for the spherical tracking shades, and on the south—37% for the awning, - 57% for the venetian, and—52% for the spherical tracking shades. The overall reduction of cooling demand of the shading systems on the east is about 10 percent lower than for the other orientations (Table 3). The energy consumed by the building system in the three shading cases in the east is similar to the two other orientations. The energy consumption of the baseline case is smaller on the east than for the other orientations, which leads to a decrease of energy consumption by the shading systems that is smaller than for the other two orientations. Overall, the annual energy demand for heating for all the orientation and shading system is very low compared to the cooling energy demand.

**Table 5. Variations of the annual lighting energy demand of the three shading systems to the baseline case.**

| | Annual lighting energy demand variation | | |
|---|---|---|---|
| Orientation | Awning | Venetian | Tracker |
| East | +57% | +81% | +84% |
| South-east | +62% | +81% | +73% |
| South | +74% | +94% | +87% |

**Table 6. Variations of the annual heating energy demand of the three shading systems to the baseline case.**

| Orientation | Annual heating energy demand variation | | |
|---|---|---|---|
| | Awning | Venetian | Tracker |
| East | +83% | +452% | +252% |
| South-east | +73% | +567% | +209% |
| South | +59% | +865% | +147% |

This significant decrease in annual energy demand is mostly due to a large reduction of the cooling energy demand for all the cases (Table 4). The lighting (Table 5) and heating energies (Table 6) both increase, but they represent a small fraction of the total energy demand. Therefore, the total energy demand is still reduced overall.

The lighting energy needed for the awning shades increases as the façade is rotated from the east (+57%) to south (+74%). Simultaneously, the cooling energy demand for the awning system decreases when moving from the east (-38%) to south (-50%). For the two other shading systems, no correlation between lighting and cooling energy demands seem to appear.

The test building is located in Princeton, NJ, with a 40° latitude and simulated with high thermal resistance materials. The low values of the heating energy demand indicate the perimeter space is cooling dominated. The annual heating energy demand represents 3% of the total annual energy on average for the four shading options, 8% on average for the venetian shades.

The total energy demand is further observed in an hourly distribution (Fig 8). The hourly energy demands higher than 1kWh are highlighted for contrast between cases.

## 4.2. Daylight conditions for the optimized positions

**4.2.1. Maximum illuminance.** All three shading systems satisfy the constraint of maximum illuminance set in the optimization system. As represented in Fig 9, no maximum

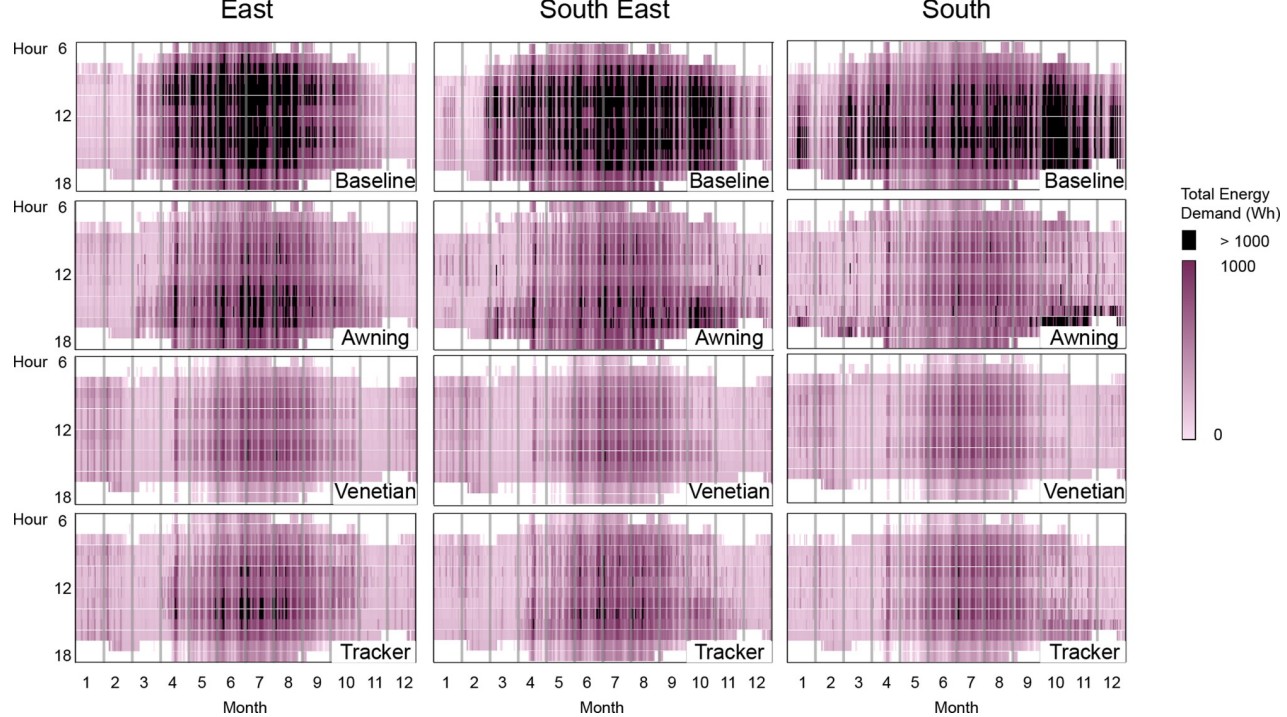

**Fig 8. Hourly total energy demand for east, south-east and south orientation and for the baseline case, awning, venetian and spherical tracker shades.** Hourly demands over 1kWh are highlighted in black.

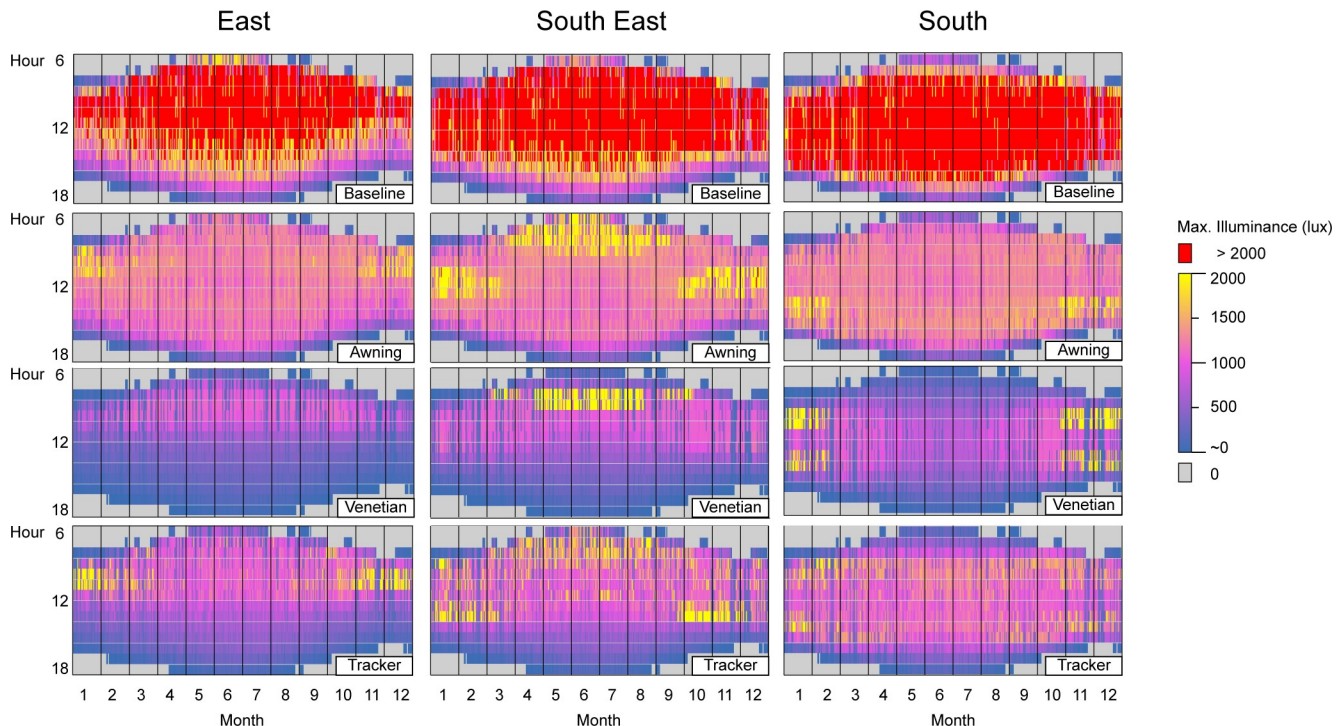

**Fig 9. Hourly maximum daylight illuminance resulting from the optimization for the east, south-east and south orientations and for the baseline case, awning, venetian, and spherical tracker shades.** Average illuminances over 2000 lx are highlighted in red.

illuminance over 2000 lx is reported for the 9 shading cases simulated. The constraint of the optimization is satisfied for all 4306 sun hours of each of the 9 cases.

The baseline case reveals patterns of hourly available daylight for each orientation (Fig 9). High illuminance values (>2000 lx) occur dominantly in the morning on the east and appear to be centered around the noon sun hour on the south. The south-east orientation is an intermediate case with both high values of illuminance in the morning and later in the day. For those three orientations, the annual mean value of maximum illuminance is well beyond the 2000 lx comfort threshold (Table 7), which signals that if unshaded this room would be subject to frequent and intense discomfort. In addition, the standard deviation of maximum illuminance is close to the mean value itself, an additional indication (if one was needed) of the extreme variability of environmental daylight.

It is therefore significant that the variability of maximum illuminance for the three shading systems is so greatly reduced. The standard deviation of the maximum illuminance is an order of magnitude lower for shaded cases than for the baseline unshaded case (Table 7). For the

**Table 7. Statistical distribution of maximal illuminance received on the work plane for the east, south-east, and south orientations and for the unshaded and shaded cases.**

| Orientation | Maximum illuminance | | | | | | | |
|---|---|---|---|---|---|---|---|---|
| | Baseline | | Awning | | Venetian | | Tracker | |
| | mean (lx) | std dev (lx) | mean (lx) | std dev (lx) | mean (lx) | std dev (lx) | mean (lx) | std dev (lx) |
| East | 2273 | 1827 | 1095 | 375 | 418 | 327 | 676 | 443 |
| South-east | 4290 | 5569 | 1176 | 577 | 572 | 469 | 805 | 521 |
| South | 3492 | 3680 | 1135 | 389 | 581 | 455 | 862 | 458 |

three orientations, the overall mean maximum illuminance is ~1100 lx for the awning, ~520 lx for the venetian, and ~780 lx for the spherical tracker shading systems. For the venetian and spherical tracker, the mean maximum illuminance is close to the 500 lx target for the work-plane average illuminance.

**4.2.2. Average illuminance.** High average values (Fig 10) coincide with the high maximum illuminance values (Fig 9) described in Section 4.2.1. Similarly, as for the maximum illuminance, the amount of daylight penetrating the space is excessive without shading. Most baseline average work-plane illuminance values are superior to 600 lx for the east (58%), for the south-east (64%) and for the south (65%) (Fig 11). The intervals below 600 lx each represent about 3% of the number of baseline values for each orientation of the façade. As shown in Fig 10, those values occur on the fringe of the high daylight periods.

The mean value of the average work-plane is 868 lx on the east, 1103 lx on the south-east and 1006 lx on the south. In addition, the variability of the unshaded average work-plane illuminance is high. The standard deviation is of the same order of magnitude as the mean value (Table 8).

The 500 lx average illuminance constraint can only be maintained in shaded cases if the unshaded baseline case provides at least 500 lx of average illuminance. None of the hours that initially provide less than 500 lx of average illuminance (deep blue in Fig 10) see an increase of illuminance once shaded.

The three shading systems meet this constraint with various levels of success. Over the three orientations, the 500 lx target is met on average 3% of annual sun hours for the baseline, 68% for the awning, 16% for the venetian, and 35% for the spherical tracking shades (Table 9). The efficiency of each shading system is relatively similar for each orientation. The success of the

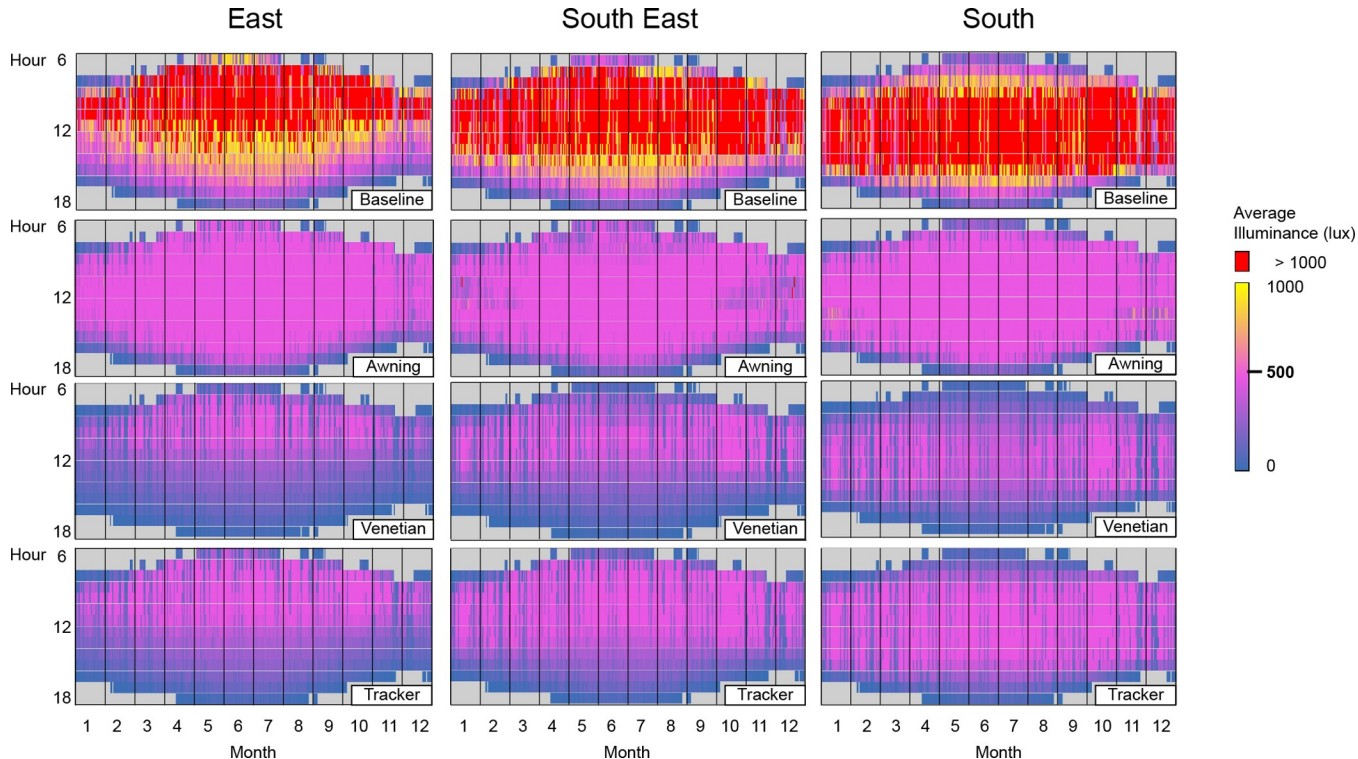

**Fig 10. Hourly average daylight illuminance resulting from the optimization for east, south-east and south orientation and for the baseline case, awning, venetian, and spherical tracker shades.** Average illuminances over 1000 lx are highlighted in red.

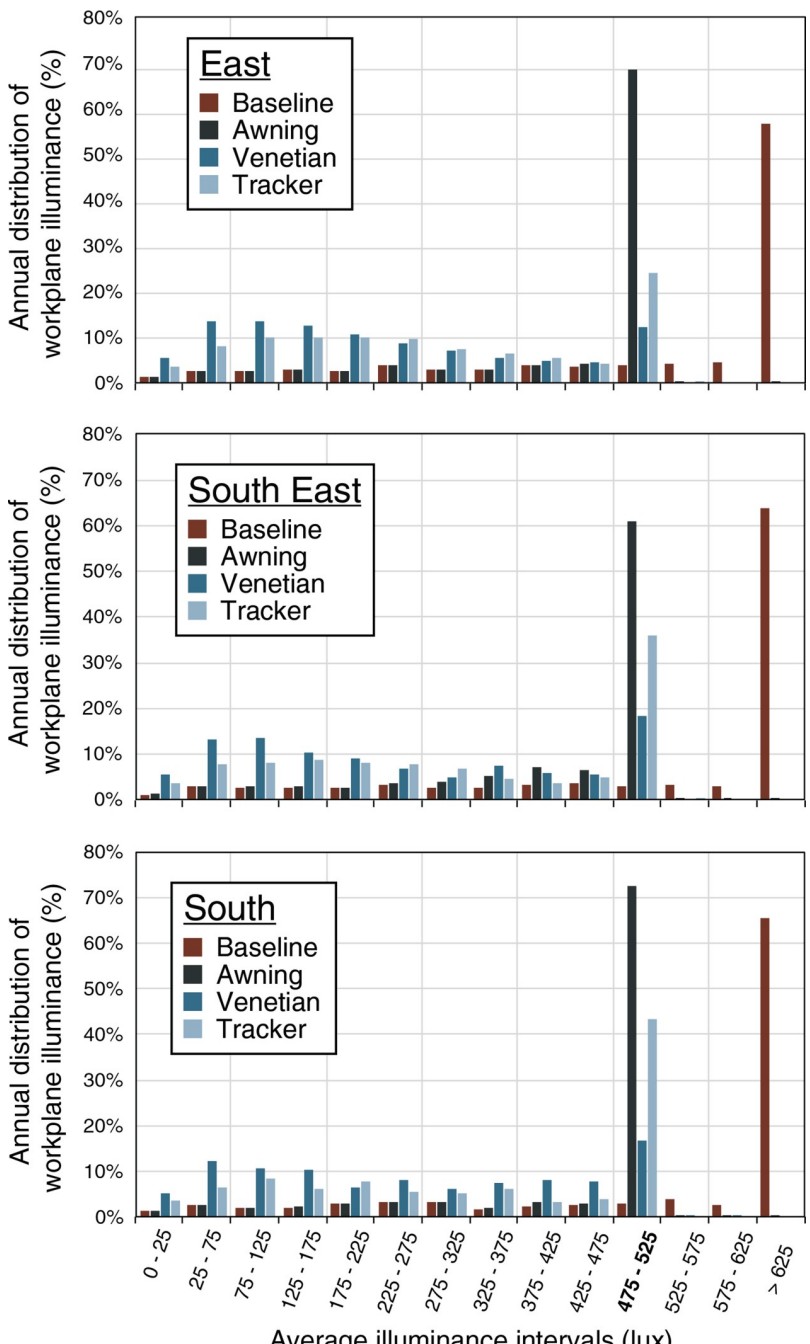

**Fig 11. Annual distribution of average illuminance occurrences for the east, south-east, and south orientations.**
The occurrences are shown in intervals of 50 lx, which correspond to +/- the 25 lx tolerance of the optimization. The
475–525 lx interval is the target interval for the optimization.

optimization for the annual hourly cases, established as reaching the expected 500 lx constraint
value consistently, translates as a uniformly colored hourly map (Fig 10) and a mean average
work-plane illuminance close to 500 lx (Table 8). The awning shades perform the best with
almost entirely uniform maps (Fig 10) and mean values of the average illuminance above 400 lx
for the three orientations. This superior performance is confirmed in Fig 11 with the awning

**Table 8. Statistical distribution of average illuminance received on the work plane for the east, south-east, and south orientations and for the unshaded and shaded cases.**

| Orientation | Average illuminance | | | | | | | |
| --- | --- | --- | --- | --- | --- | --- | --- | --- |
| | Baseline | | Awning | | Venetian | | Tracker | |
| | mean (lx) | std dev (lx) | mean (lx) | std dev (lx) | mean (lx) | std dev (lx) | mean (lx) | std dev (lx) |
| East | 868 | 642 | 427 | 135 | 227 | 155 | 283 | 164 |
| South-east | 1103 | 879 | 418 | 143 | 251 | 169 | 314 | 173 |
| South | 1006 | 666 | 432 | 135 | 266 | 167 | 336 | 174 |

shades only failing when the daylight does not provide 500 lx in the baseline case. In contrast, the spherical tracking and venetian shading systems do not perform as well. The annual hourly map is less uniform than for the awning shades, and the mean values of the illuminance are contained in the [220, 330] lux interval. This is confirmed in Fig 11 by the small percentage of the successful cases for the venetian shades and to a lesser extent for the spherical tracker.

The three shading systems have 0.1% of occurrences in the intervals above 525 lx. The optimization methodology implemented successfully limits the amount of daylight transmitted by the shades into the space.

## 4.3. Use of each degree-of-freedom

The position optimization also enables the analysis of the shades kinematic systems. In particular, the results indicate the variability of use of each degree-of-freedom for the tested shading systems. The awning and the venetian shades are systems with a single degree-of-freedom while the spherical tracker is a dual degree-of-freedom system. The range of actuation goes from 0 to 1 for all the degrees-of-freedom (see Section 3.2 for detail of the case study's kinematics). The awning is fully open at initial position (dof1 = 0) and fully obstructing at final position (dof1 = 1). The venetian shades are fully closed at initial position (dof1 = 0) and fully open at final position (dof1 = 1). The spherical tracker's elevation is closed in initial position (dof1 = 0) and fully open in final position (dof1 = 1), while the azimuth is at a -45˚ angle at initial position (dof2 = 0) and at a +45˚ at final position (dof2 = 1). Table 10 shows the annual standard deviations and averages for the the d.o.f. of each shading system.

The standard deviation $\sigma$ is an indicator of how often the d.o.f. is activated. If a d.o.f. is never activated, $\sigma$ will be close or equal to 0. For the three orientations, $\sigma$ has similar values across shading systems (Table 10). $\sigma_{Awning}$ and $\sigma_{DoF2\_Tracker}$ are the highest values of standard deviations for the case study. With values around or above 0.3, they indicate that these degrees-of-freedom have a good variation in their use. In contrast, $\sigma_{Venetian}$ and $\sigma_{DoF1\_Tracker}$ are low and indicate that the d.o.f. operate around the mean.

The average value of a d.o.f. provides information on the average position of the shading system. For each one of the shading systems tested in the case study, the average position is

**Table 9. Success rate of the optimization for the 500 lx average illuminance constraint.** The baseline case is not part of the optimization and is given as an indicator of the unshaded situation. The values represent the frequency over the 4306 annual sun hours that the average work plane illuminance will be in the 475–525 lux range.

| Frequency of average work plane daylight quantity satisfying the 500 lx constraint (% of annual sun hours) | | | | |
| --- | --- | --- | --- | --- |
| Orientation | Baseline | Awning | Venetian | Tracker |
| East | 4% | 70% | 12% | 25% |
| South-east | 3% | 61% | 18% | 36% |
| South | 3% | 72% | 17% | 43% |
| **Average** | 3% | 68% | 16% | 35% |

**Table 10. Annual standard deviation and average for the d.o.f. of each shading systems of the case study.**

| | Annual d.o.f. Standard deviation | | | |
|---|---|---|---|---|
| Orientation | Awning | Venetian | Tracker 1 | Tracker 2 |
| East | 0.270 | 0.057 | 0.118 | 0.399 |
| South east | 0.316 | 0.110 | 0.199 | 0.367 |
| South | 0.291 | 0.118 | 0.197 | 0.388 |
| | Annual d.o.f. Average | | | |
| Orientation | Awning | Venetian | Tracker 1 | Tracker 2 |
| East | 0.322 | 0.984 | 0.953 | 0.645 |
| South east | 0.392 | 0.955 | 0.899 | 0.704 |
| South | 0.386 | 0.956 | 0.892 | 0.648 |

similar on each orientation (Table 10). For the awning and the second d.o.f. of the spherical tracker, that average is well into the 0.3 to 0.7 range of the actuator range. This supports the previous result that throughout the year those d.o.f. are active because the average value +/- the standard deviation covers much of the 0 to 1 total range of the actuator. In opposition, the venetian shades and the first d.o.f. of the spherical tracker have average values close to 1. The most common position for these d.o.f. is to be fully opened. Combined with a low standard deviation for these d.o.f., these average values express the quasi fixed nature of these d.o.f.. This immobility in turns suggest an issue with the kinematic design of both shading systems. The average position of those tow d.o.f. is to be fully opened. An open position would let the most daylight in the room. From Section 4.2, it appeared that while the venetian shades and the spherical tracker had good thermal performance, they were consistently unable to provide sufficient amounts of daylight in the test room. The density plot of the values of actuation for each d.o.f. (Fig 12) provides a detailed view of the annual distribution for the values in Table 10. The venetian and first d.o.f. of the tracker have a high density for around 1. For these

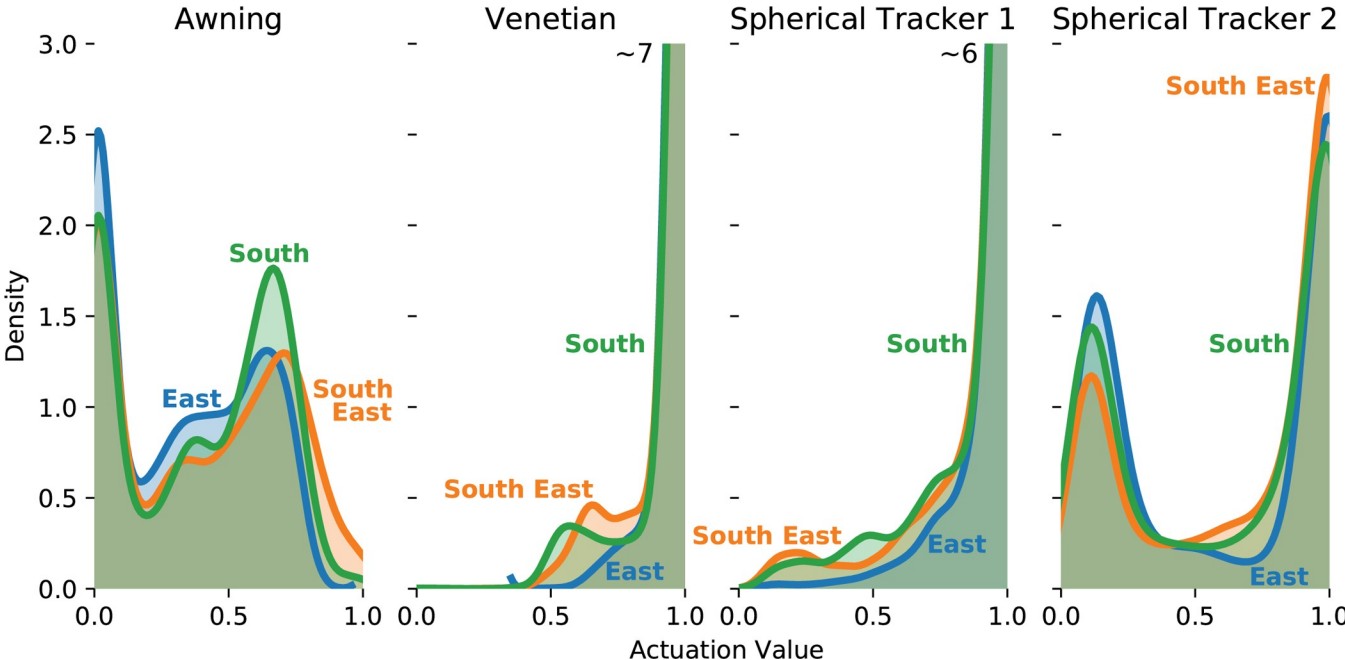

**Fig 12. Density plots of the actuation for each shading system and their respective d.o.f.** The bandwidth of the plots is 0.05 (in units of actuation).

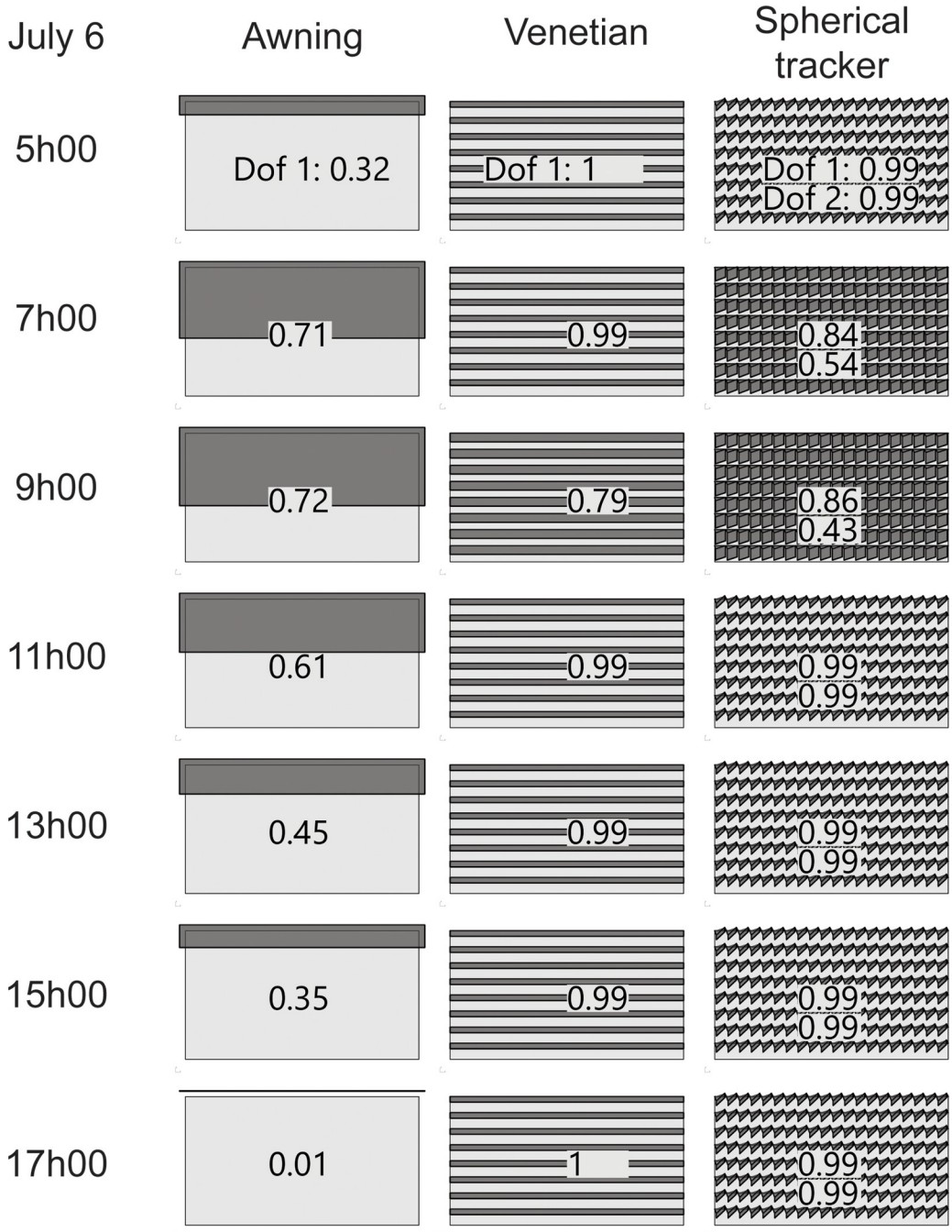

**Fig 13.** The degree-of-freedom of the awning shades ($\sigma_{July\ 6} = 0.24$) is more often activated than the venetian shades ($\sigma_{July\ 6} = 0.06$) and the spherical tracker ($\sigma_{dof\ 1-July\ 6} = 0.11$, $\sigma_{dof\ 2-July\ 6} = 0.31$).

d.o.f. the tendency of the values to be skewed towards one value indicate their lack of usefulness. The result of insufficient daylight levels from Section 4.2 is supported by the kinematic study. An example of kinematic sequence for July 6 (Fig 13) exemplifies the lack of motion for the venetian shades and to a smaller extend to the spherical tracker.

## 5. Discussions

In all presented cases, the governing assumption is that user comfort should be the controlling parameter to a dynamic shading system. Mathematically, this translates to setting the energy demand parameters (heating, cooling and lighting) as the objective to be minimized and the visual comfort as constraints to be met. This methodology is adaptable since the choice of criteria for the objective or the constraints is selected by the designer. The results presented in this study are specific to the case chosen at 40° latitude and east to south building orientations. They are representative of a design case one might encounter. The methodology, however, can be applied to many cases. It can be used in early design phases as well as for in-depth simulations and shading system design. The methodology will find use in early design phases of façade systems because it allows one to compare several design cases for a specific environmental context. It can also be used to refine the design of a specific shading system by quantifying effects of parametric variations on the system's performance.

### 5.1. Performance of the methodology

The total run-time of the analysis was 73 minutes on an Intel i7 4 core CPU at 4.20GHz. This total time is decomposed in 21 minutes for the daylighting analysis in Radiance, 42 minutes for the thermal analysis in EnergyPlus and 10 minutes for the optimization in MATLAB. The daylight and thermal analyses are parallelized on 8 processes. The optimization is parallelized in 4 processes. The simulation of the building physics takes much of the analysis time. Running the simulations independently from the optimization allows to run the building physics simulations as fast as the machine tolerates and it allows to store the results for later use. For a given case study, once the simulations have been performed, the optimization can be run for any value of the illuminance constraint without having to perform the simulations again. The decision to reduce the number of calculations (Section 2.3) allows to keep the thermal simulation short enough that the methodology can be iterated quickly. In comparison, the optimization of the shading system is fast. The choice of genetic algorithm for global minimum method shows to be appropriate. The maximum number of generations for the genetic algorithm solver is set to 15 and the algorithm converges in 3 to 4 generations for most solar positions. This method also allows the designer to easily and quickly visualize the impact of their comfort constraints decisions on the performance of the system. The values for the explicit constraints (max and average daylight quantity in the case study) can be adjusted at the optimization stage. This feature lets the decision maker adjust the comfort requirements of the systems.

As mentioned in Section 2.5, thermal lag is not explicitly included in the system. Thermal lag is the impact of a shading condition at time t on the following time steps. Alternatively, it is the impact of the prior timesteps on the current timestep t. In our case, the system is does not reset to initial conditions at each of the 4306 sun hours. The thermal lag is therefore included (as explained in Section 2.5) due to the way the simulations are performed. Each position of a shading system is treated as a fixed shade for simulation. The interpolation recreates the dynamic behavior. There is no impact of this method on the illuminance or on the lighting energy measurements since daylight is an instantaneous parameter. The heating and cooling energies might however be impacted, due to thermal lag. For these two parameters, the heating and cooling energy values implicitly include the lag of previous timesteps.

### 5.2. Comparison of shading devices in the methodology

Providing improved daylight access can be a barrier to energy performance. There is a necessary compromise between reducing the thermal input due to solar radiation and providing daylight for occupants. The awning shades exemplify this compromise. They perform best on

meeting the daylight constraints but necessitate more heating and cooling energy to maintain thermal comfort than the other shading systems. All three shading systems limit the maximum illuminance to the 2000 lx set in the optimization system. However, only the awning shading system delivers the desired average work-plane illuminance conditions consistently. For 68% of the annual sun hours (16% for venetian shades and 35% for spherical tracker) the average illuminance equals the 500 lx constraint. This superior daylight performance is balanced by a lower reduction of annual total energy demand compared the two other shading systems (Section 4.1).

Regarding which shading system to choose for the study of this paper, the awning performs the best visually, but long overhanging awnings are more sensitive to façade wind loads compared to an external venetian shading system with 20 cm wide slats. If the durability of the shading system is a concern, using textile material in the awning might not be the best choice. Those additional criteria would help refine the choice of parameters for the design of a shading system. For the awning shades, the east and south orientations present the best overall results. The south-east orientation is more difficult to tackle with this type of shade. This reinforces the known difficulty of treating south-east orientations due to the solar vectors' high incident angles with windows. In that orientation, both results of the daylight and the energy demand are worse with the awning.

For all cases and all orientations, the annual energy demand presents a very low energy demand for heating. Several factors explain this. With a large value of 65% of WWR, this result indicates that solar radiations play a significant role in heating the test room. In addition, the orientations included in this study (east, south-east and south) receive solar radiations every day for a large portion of the day (see Fig 8). Third, the R-value for the window (13.3 K·m$^2$/W, default value in DIVA) is larger than average. In buildings, windows are usually the weak link of thermal insulation, in this study that is not the case. The baseline case receives a lot of solar radiation: it has the lowest amount of heating demand and the highest demand for cooling. This confirm that the prevention of overheating due to solar radiation is the controlling mode of operation of the test room.

## 5.3. Design shortcomings and goals for dynamic shading systems

The inability of the venetian and spherical tracking shading cases to meet the daylight constraint of average illuminance is due to the design of the shading systems themselves. Those two systems cannot be stored away and always obstruct the view to some degree. In Section 4.3, the analysis of the kinematics validates this shortcoming of the proposed shading systems by showing that the average position for those shades was fully open. And that even by being fully opened they failed to provide the 500 lx of average illuminance requested by the constraints of the optimization. Therefore, as described in Section 4.2.2., the daylight provided by the baseline case will always be decreased by those two shading systems. The proposed optimization methodology allows for the identification of those deficiencies and allows for an iterative redesign approach to take place. Once the shortcomings of a given shading system have been identified (such as the need for fully unobstructed window) the design can be modified and analyzed by the methodology one more time.

Judging whether a system has reached its optimal kinematic design is difficult. Since solar shading is a design problem, there is no single right answer. Therefore, any solution proposed should come from a thorough iterative and comparative process. This solution should be specific to the building and specific to the latitude of the building. As was showed in this case study, a degree of freedom that is regularly activated during the year (Fig 12) is a sign that the kinematic system is performing well. However, if a degree of freedom is not activated in the

optimization framework, the kinematic system will not be able to provide the desired performance. In the case study the awning and the second d.o.f. of the spherical tracker are the most efficient d.o.f.. A further iteration of the shading system should try to combine both kinematic properties and be evaluated by the methodology.

## 5.4. On the choice of constraint values

The choice of comfort criteria to be used as input in the optimization system influences the outcome of the analysis. A choice of a lower average work-plane illuminance constraint may have produced different results, for instance. The venetian shadings and the spherical tracking shades produce average illuminances consistently in the interval 220 lx-330 lx in our study. Setting the constraint within that range could have shown one of these two shading systems as the best overall system. However, this would have certainly obstructed the fact that those are less versatile than the awning system in their tested configurations. They provide low average daylight quantities because they do not allow unobstructed views to the outside and therefore are limited in the amount of daylight they can let in diffuse light situations. Similarly, the spherical tracking shades are intuitively better than venetian shades for unobstructed views from the inside to the outside of the space, due to their ability to twist out of plane. Overall however, the awning shades are the best performer of the three types of shades due to their ability to be fully openable. The present methodology can help refine the design of this shading system. In their present configuration, the awning shades will most likely need a lot of maintenance due to being designed as a fabric dynamic system. This system is very susceptible to wind and rain, which make it unrealistic to practical implementation. Finally, their energy performance can be improved to match what the other two systems provide.

Setting adequate constraints for an optimization procedure will shape the type of shading that will perform best. Picking the right constraints is therefore essential. In this methodology, the constraints are based on human comfort, hence occupants are put at the center of the study, in a position that promises to increase wellbeing and enjoyment in the building.

## 6. Conclusion and future work

This study introduces a methodology for the evaluation and optimization of the performance of shading systems to reduce heating, cooling, and lighting energy demands, and maintain the visual comfort of user into acceptable ranges. This analysis opens a path to fulfill the currently underdeveloped potential for holistic performance improvement in façade design and substantiates the benefit of more advanced shading systems. The methodology is based on the interpolation of simulation results for the number of actuation states of the shades and the minimization of objective functions under constraints. Three types of dynamic external shades have been analyzed in a case study and compared with the aim of reducing the solar gains on a building office glazing while maintaining a precise work plane illuminance level. With the metrics chosen (daylight quantity and electric energy), the three shading systems were analyzed kinematically. Two degree-of-freedom were identified as showing usefulness. The analysis led to the conclusion that two of the shading devices tested were not able to provide the necessary performance due to their kinematic design. This study also confirms the result of previous studies that dynamic shading decreases the cooling loads dramatically but tends to increase the heating and lighting loads.

The implementation for the control of existing dynamic shade is possible with this methodology. Once the methodology has been implemented to refine the design of shading system, this new solution must be operated in the actual building. The presented methodology is not a control algorithm for dynamic shading, but it provides a path for the development of more

capable shading systems. However, the methodology can be adapted to be an operation-oriented algorithm. This change to the methodology can be carried in a two-step process. For a given building and room, the interpolated function should be trained for all the possible weather types (e.g. clear, mixed or overcast sky). The optimization should then be run at each time step by selecting the instantaneous weather condition. This would therefore require that sensors determine the weather condition at the current time and that a computing unit ran the optimization system with the correct set of interpolated functions. The issue with this operation method is that it requires knowledge of the building's floor plan and have access to geometric models compatible with EnergyPlus and Radiance. Future work should focus on circumventing this limitation of the operation while still be able to access the high multi-dimensional capabilities that using this design methodology provides.

The critical aspect of this methodology for use in practice is choosing the appropriate metrics of performance and comfort adapted to the environmental context of the building. This selection of metrics must be done by considering the comfort of occupants. It is the root of the methodology. The occupants are central to the physical design of the dynamic shading system. By providing actionable feedback on the physical design of a dynamic shading typology by centering it on the comfort of occupants, the design space of novel facade systems gains another dimension. This opens the possibility realizing the promise of improved building environments that dynamic shading systems embody but never have realized on a large scale.

## Supporting information

**S1 File. Case study simulations.** Case study data including heating, cooling and lighting energy, average and maximum work plane illuminance for the three shading systems and for the baseline case. Data presented in database file format.
(ZIP)

**S2 File. Case study results.** Optimization output data including optimal positions, energy demands and produced work plane illuminance. Data presented in database file format.
(ZIP)

## Acknowledgments

Many thanks to Georgina Hall from INSEAD Decision Sciences Department for her advice on interpolation.

## Author Contributions

**Conceptualization:** Victor Charpentier, Forrest Meggers, Sigrid Adriaenssens, Olivier Baverel.

**Data curation:** Victor Charpentier.

**Formal analysis:** Victor Charpentier.

**Funding acquisition:** Sigrid Adriaenssens.

**Investigation:** Victor Charpentier.

**Methodology:** Victor Charpentier.

**Project administration:** Victor Charpentier.

**Software:** Victor Charpentier.

**Supervision:** Forrest Meggers, Sigrid Adriaenssens, Olivier Baverel.

**Validation:** Victor Charpentier.

**Visualization:** Victor Charpentier.

**Writing – original draft:** Victor Charpentier.

**Writing – review & editing:** Victor Charpentier, Forrest Meggers, Sigrid Adriaenssens.

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
