## [Decision Letter · Decision Letter 0]

23 Dec 2019

PONE-D-19-30597

Non-deterministic control of external dynamic shading

PLOS ONE

Dear Dr. Charpentier,

Thank you for submitting your manuscript to PLOS ONE. After careful consideration, we feel that it has merit but does not fully meet PLOS ONE’s publication criteria as it currently stands. Therefore, we invite you to submit a revised version of the manuscript that addresses the points raised during the review process.

We would appreciate receiving your revised manuscript by Feb 04 2020 11:59PM. To enhance the reproducibility of your results, we recommend that if applicable you deposit your laboratory protocols in protocols.io, where a protocol can be assigned its own identifier (DOI) such that it can be cited independently in the future. For instructions see: http://journals.plos.org/plosone/s/submission-guidelines#loc-laboratory-protocols

We look forward to receiving your revised manuscript.

Best regards,

Marco Lepidi, Ph.D.

Academic Editor PLOS ONE

Additional Editor Comments:

Dear Authors,

two detailed reports have been collected for your submission. Both the reports are encouraging, but recommend major improvements.

Please revise your manuscript according to the critical points and/or suggestions raised up by the reviewers.

Best regards

Marco Lepidi

Reviewers' comments:

Reviewer's Responses to Questions

**Comments to the Author**

1. Is the manuscript technically sound, and do the data support the conclusions?

Reviewer #1: Partly

Reviewer #2: Yes

2. Has the statistical analysis been performed appropriately and rigorously? 

Reviewer #1: N/A

Reviewer #2: Yes

3. Have the authors made all data underlying the findings in their manuscript fully available?

Reviewer #1: Yes

Reviewer #2: Yes

4. Is the manuscript presented in an intelligible fashion and written in standard English?

Reviewer #1: Yes

Reviewer #2: Yes

5. Review Comments to the Author

Reviewer #1: Investigations on the operation of dynamic shading devices are important nowadays considering the more and more widespread use of such technology in buildings. The authors describe a methodology for the control of dynamic shading devices to minimize energy use while guaranteeing the visual comfort of occupants, as well as its application for the evaluation of three shading systems for thee orientations (east, south east and south).

The authors are encouraged to address the following issues:

1. General comment: considering that the focus of the paper is primarily on the description of a methodology for the evaluation of the performance of dynamic shading devices, it is suggested to revise the organization of the manuscript and of its sections. For example, the methodology could be described first, while its application to the selected shading options later. Similarly, it is suggested to change the title to better highlight the scope of the paper. The wording “non-deterministic” present in the title is not reported in the text of the manuscript and therefore is not properly explained.

2. General comment: The comparison with a baseline is appreciated and required to understand the value of the proposed solution. However, it would be more useful if the baseline was not an unshaded window (despite it is recognized its widespread presence in buildings in the USA), but the same types of blinds with a different and “more common” control algorithm, considering that the control algorithm suggested is the novelty of this paper. It is suggested to expand the comparison to a different (or additional) baseline model.

3. Abstract, line 39: it is surprising to read that electricity is used to maintain the constraint about daylight level. It is assumed that the constraint is about light level and not daylight level, considering that daylight is not linked to energy use.

4. Abstract: why only results associated to the spherical solar tracking are presented in the abstract? Is the primary goal of the paper to evaluate a specific shading device? It is suggested to reformulate the abstract to be focused on the actual goal of the paper.

5. Lines 63-64: the current wording of this sentence is not very clear. What did the authors mean with it?

6. Line 136: remove electric lighting from the sentence as it does not affect the thermal comfort (unless radiation from operating electric light is included into the modeling?).

7. Lines 162-163: this information has already been presented. Please describe the simulation model in one section of the manuscript only (it is suggested to dedicate to it a specific subsection).

8. Lines 173-177: revise the caption of the figure.

9. Line 189: the authors indicate to have modeled three blind solutions that are similar to commercially available systems. However, especially for the modeling of the venetian blinds, it appears that too many simplifications were done. In the commercially available system indicated by the authors, for example, the slats are not flat but present more elaborated geometries. In addition, a depth of 27.5 cm (line 206) is considered too big compared to commercially available systems. Why did the authors modeled the slats in this way? Why not choosing less deep slats and increasing the umber of slats to cover the height of the window? Also the dimensions of the spherical solar tracker components are pretty big. In practice, this could cause problems with the mechanical systems of the actuators.

10. Lines 210-212: the authors might consider modifying the text in this part of the manuscript to indicate that the awning is the only system that allows to have a completely unobstructed view to the outside. The other two systems allow to totally close the window, but the current text might confuse the reader as it seems that only the awning allows it.

11. Line 234: please correct the error in the subtitle of the text (now in line with the previous paragraph).

12. Lines 239-240: please indicate the heating and cooling temperature set-backs.

13. Lines 241-244: If the solar radiation calculation does not consider the daylight simulations (hence, it is guessed also the position of the shading devices to guarantee the minimum and maximum illuminance levels at the workplane), do energy results include the dynamic positions of the shading systems? If not, how can energy results be different? If yes, how is the information related to the shading device integrated into the energy simulations? It is suggested to the authors to better explain this process to avoid confusing the readers.

14. Lines 254-255: please correct the units.

15. Line 266: the use of daylight illuminance as a metric for measure daylight quality is not appropriate. Daylight quality also refers to distribution, color, dynamics. Please refer only to daylight quantity.

16. Line 266: if there is a sensor for each square in figure 1, what are the four sensors indicated in the figure?

17. Lines 272-273: why is the information about window transmittance repeated?

18. Lines 296-297: why dividing the space in 90x90 cm grid if only 4 sensors are used? What is the use of the remaining “squares”?

19. Line 302: from the description of the text it is argued that the artificial light is always on despite daylight provides the minimum illuminance on the desk plane? Is this correct? If yes, why not turning off the electric light?

20. Lines 304-306: what do the authors mean with this sentence? That daylight diminishes with the depth of the room? Please clarify. In addition, why two lighting systems (e.g., front and back of the room) were not modeled? This would have saved electric light energy as occupants close to the window would have probably had the necessary illuminance level from daylight alone.

21. Lines 307-311: the explanation provided is somehow confusing. Are the authors saying that only energy demand for electric light is calculated, but not the final illuminance (natural + artificial) on the work plane? Please clarify the text.

22. Line 412: glare is included into the visual comfort.

23. Section 2.5-2.6: It is not clear how the optimization process works: considering the two constrains (visual and thermal comfort), which is prioritized? Is the information related to visual comfort (more light and more solar radiation) integrated into the thermal simulation?

24. Line 419: how is the validation performed? Please explain it in detail for reproducibility.

25. Lines 441-442: in reality, the energy consumed by the three shading systems in the east orientation is very similar to the other two orientations. The decrease of energy consumption results smaller as the energy consumption of the base case on the east orientation is already lower compared to that of the same case in other orientations.

26. Line 444: the figure appears to be in kWh and not in MJ.

27. Figure 6: it is surprising to see that the annual energy demand for heating for all the orientations and shading systems is very low at the latitude of Princeton. Can the authors explain this result? Was the room heated by the incoming solar radiation? If yes, was the solar radiation changed according to the operation of the shading devices?

28. Figure 7: the total energy demand is characterized by the cooling needs. However, is it surprising to see that for the awning and the east orientation model, the higher energy demand occurs in the afternoon. How can this result be explained? By the thermal accumulation of indoor surfaces?

29. Figure 8: please add illuminance values to the colored legend to make the graph easier to understand.

30. Line 502: are the values displayed only related to daylight? What about the fact that the electric light was always on?

31. Line 511: reference to figure 7 is wrong.

32. Lines 640-642: can the authors expand on this topic? Wouldn’t the awning be the best option in terms of quantity of view out as it is the only one fully openable?

33. Line 652: it is suggested to use the wording “and maintain the visual comfort of user into acceptable ranges”.

34. Table 9: it is not clear how the percentages are calculated. Please explain in the text or in the caption of the table.

35. Conclusions: could the authors better explain how researchers and practitioners can apply the suggested methodology in practice? In a real building, how would the suggested control algorithm work?

Reviewer #2: The paper investigated the Genetic Algorithm to optimize external dynamic shading in the USA. The study is rich in content and explanation. Generally, the manuscript is well written and the research novelty is linked to the development of dynamic optimization method. However, there are some comments should be considered to improve the research quality:

1- The abstract mentioned

"The heating, cooling, artificial lighting and daylight (work plane illuminance) performances are evaluated with Radiance and EnergyPlus based on local weather data."

However, the results mainly focus on daylight analysis while heating, cooling, artificial lighting are superficially covered. A proper balance among different factors should be provided.

2- The paper should improve its literature with the latest research to support the research decisions, for example:

The study mentioned that the set point of the lighting system is 500 lx without providing an explanation. Also, the study mentioned "The simulated perimeter office is 5 m deep, 4.5 m wide, and 3.2 m high" without providing support from the literature. Including the study below would help to support this claim.

Luo, Y., Zhang, L., Su, X., Liu, Z., Lian, J., & Luo, Y. (2019). Improved thermal-electrical-optical model and performance assessment of a PV-blind embedded glazing façade system with complex shading effects. Applied Energy, 255, 113896.

Al-Obaidi, K. M., Munaaim, M. A. C., Ismail, M. A., & Rahman, A. M. A. (2017). Designing an integrated daylighting system for deep-plan spaces in Malaysian low-rise buildings. Solar Energy, 149, 85-101.

Also, the study mentioned about, the algorithm for electric lighting is open-loop; it does not feedback into the daylighting assessment. However, the study does not discuss different feedback systems as open-loop or closed-loop in dynamic shading systems. Including the study below would help to support this claim.

Al-Masrani, S. M., & Al-Obaidi, K. M. (2019). Dynamic shading systems: A review of design parameters, platforms and evaluation strategies. Automation in construction, 102, 195-216.

Ayoub, M. (2018). Integrating illuminance and energy evaluations of cellular automata controlled dynamic shading system using new hourly-based metrics. Solar Energy, 170, 336-351.

3- The study needs to explain the outdoor daylight and sky conditions in Mercer County, New Jersey, USA that would control the study outcomes.

4- The study mentioned about 4 sensors as shown in Figure 1 but I couldn't see the results of each sensor.

5- The study mentioned ""The total run-time of the analysis was 73 minutes on an Intel i7 4 core CPU at 4.20GHz. This total time is decomposed in 21 minutes for the daylighting analysis in Radiance, 42 minutes for the thermal analysis in EnergyPlus and 10 minutes for the optimization in MATLAB."

How could this statement help the outcomes of this research?

6- Figure 2, model C, it is unclear to understand the model movement and rotation.

6. PLOS authors have the option to publish the peer review history of their article (what does this mean?). If published, this will include your full peer review and any attached files.

Reviewer #1: No

Reviewer #2: No

---

## [Author Response · Author response to Decision Letter 0]

17 Feb 2020

Manuscript PONE-D-19-30597

Answer to Reviewer #1

In this document, the authors address the comments of reviewer 1. The authors would like to thank reviewer 1 for his/her thorough and very helpful comments aimed at improving the quality of the manuscript. We are grateful for the time and effort spend by reviewer 1. We hope the changes we have made to the manuscript will be satisfactory as we believe the resulting manuscript is much improved.

General comment:

Reviewer #1: Investigations on the operation of dynamic shading devices are important nowadays considering the more and more widespread use of such technology in buildings. The authors describe a methodology for the control of dynamic shading devices to minimize energy use while guaranteeing the visual comfort of occupants, as well as its application for the evaluation of three shading systems for thee orientations (east, south east and south).

Detailed comments:

1.General comment: considering that the focus of the paper is primarily on the description of a methodology for the evaluation of the performance of dynamic shading devices, it is suggested to revise the organization of the manuscript and of its sections. For example, the methodology could be described first, while its application to the selected shading options later. Similarly, it is suggested to change the title to better highlight the scope of the paper. The wording “non-deterministic” present in the title is not reported in the text of the manuscript and therefore is not properly explained.

The authors would like to thank the reviewer for the challenging comment. As a result, two major changes have been brought to the paper. 

First the title of the manuscript was changed to “Occupant-centered optimization framework to evaluate and design new dynamic shading typologies”. The authors think it now better reflects the contribution of the paper which is to evaluate and compare the performance of dynamic shading systems.

Second, the manuscript was reorganized as suggested by the reviewer. The initial methodology section that contained both the detail of the case study and the detail of the methodology was split in two sections. The presentation of the methodology comes first in section 2: “Methodology to evaluate and compare dynamic shading typologies”. The detail of the case study comes second in section 3: “detail of the case study”. The result of this modification is that the reader can now clearly understand the methodology independent of the details of the case study. 

2. General comment: The comparison with a baseline is appreciated and required to understand the value of the proposed solution. However, it would be more useful if the baseline was not an unshaded window (despite it is recognized its widespread presence in buildings in the USA), but the same types of blinds with a different and “more common” control algorithm, considering that the control algorithm suggested is the novelty of this paper. It is suggested to expand the comparison to a different (or additional) baseline model.

The authors agree with the comment of the reviewer that in the case of the presentation of a new control algorithm a suitable baseline would be an algorithm used in practice. However, the paper is aimed at being a comparison between different typologies of dynamic shading systems. By re-organizing the manuscript, the authors additionally made the focus of the paper more precise. The paper now focuses on comparing the performances of shading systems to enable increased creativity in their design. An additional section (4.3 use of each degree of freedom) was added to make this refocus clearer to the reader. The new section provides a new way to look at the design of a shading system by allowing to visualize the use of each degree-of-freedom of the shading typologies. A high usage, i.e. frequent motion, is correlated to a highly efficient shading system while a low usage translates an issue with the design of the shading system.

The unshaded window is used as a baseline in this case because it provides a representation of the most common scenario for daylighting. In this refocused paper the goal is not to provide a control algorithm but to substantiate the usefulness of the combination of higher expectations and advanced tools for the design of dynamic shading typologies. 

3. Abstract, line 39: it is surprising to read that electricity is used to maintain the constraint about daylight level. It is assumed that the constraint is about light level and not daylight level, considering that daylight is not linked to energy use.

Thank you for pointing out this imprecision. The overall lighting level is adjusted by the electric lighting, the daylight level is fixed by environmental conditions. 

4. Abstract: why only results associated to the spherical solar tracking are presented in the abstract? Is the primary goal of the paper to evaluate a specific shading device? It is suggested to reformulate the abstract to be focused on the actual goal of the paper.

The goal of the paper is to present the methodology and to display the results of its implementation for standard and novel dynamic shading systems. The reviewer is right in pointing the inconsistency of the abstract. The last part of the abstract was therefore reformulated to reflect the broader goal of the paper. It previously was: 

“When tested on a prototype design of spherical tracking dynamic external shades, our methodology led to a 62% decrease in annual cooling energy demand, but an 81% increase in lighting energy to maintain the constraint daylight levels. Overall this shading system reduced the annual energy demand by 45% on average over the three orientations tested.”

And now reads as: 

“Over the three shading typologies tested, the average energy consumption is reduced by 44%. The tradeoff appears with a decrease of average annual cooling of 60% associated with an average increase in electric lighting of 77%. The methodology highlights the awning shades presented in the paper as the best overall performer with a good ability to control natural daylight and an overall reduction of 33% of the annual energy consumption over the three orientations tested.”

5. Lines 63-64: the current wording of this sentence is not very clear. What did the authors mean with it?

Thank you for pointing this sentence’s lack of clarity. The sentence was meant to summarize the significance of using a holistic approach to solar shading that would revolve around the improvement of the overall comfort of occupants. Psychological aspects are key to individual comfort and the proposed methodology with its flexibility of constraints can be used to integrate these notions into a formal framework. The sentence was modified and now reads: “The parameters of individual comfort are interdependent due to psychological factors. There is therefore great significance in using a holistic approach to solar shading revolving around the improvement of the overall comfort of occupants”. 

An additional sentence referring to psychological ties and the advantages of the methodology was added in the introduction: “In addition, since psychological aspects are key to individual comfort, the proposed methodology can be improved to integrate new metrics of performance.

6. Line 136: remove electric lighting from the sentence as it does not affect the thermal comfort (unless radiation from operating electric light is included into the modeling?).

The aspects of comfort under study in this paper are thermal and lighting comfort. The energy in question here is the overall consumption of the building system for to maintain a global occupant comfort level. The sentence was therefore modified to make this clearer:

“Energy demand for heating, cooling and electric lighting are calculated to maintain a fixed level of occupant comfort (thermal and lighting).” 

7. Lines 162-163: this information has already been presented. Please describe the simulation model in one section of the manuscript only (it is suggested to dedicate to it a specific subsection).

Thank you for pointing this redundancy. The first mention of the room size was removed in the introduction to section 2 detailing the content of the section.

8. Lines 173-177: revise the caption of the figure.

The caption was revised and now reads:

“Figure 1 (a) Perimeter office space – window (ww) in blue, work plane (wp) in yellow - exterior / interior ground (grd), walls (wll) and ceiling indicated – (b) daylighting grid with a daylight sensor per square, four artificial lighting sensors determine when lighting is needed”

9. Line 189: the authors indicate to have modeled three blind solutions that are similar to commercially available systems. However, especially for the modeling of the venetian blinds, it appears that too many simplifications were done. In the commercially available system indicated by the authors, for example, the slats are not flat but present more elaborated geometries. In addition, a depth of 27.5 cm (line 206) is considered too big compared to commercially available systems. Why did the authors model the slats in this way? Why not choosing less deep slats and increasing the number of slats to cover the height of the window? Also, the dimensions of the spherical solar tracker components are pretty big. In practice, this could cause problems with the mechanical systems of the actuators.

The size of the spherical tracker corresponds to the prototyped size of the novel shading system currently being developed by the authors. The system is based on active bending of shell structures and is actuated using shape memory alloy wires. In the authors’ prototype, the modules are between 25 and 30 cm tall and presented on the façade as shown in figure 2 of the initial submission. The choice of 27.5 cm corresponds to a division by 8 of the vertical height of the window. The initial manuscript mentions the relationship with the ongoing work of the authors at L.200-201. 

The size of the spherical tracker and of the venetian shades are related. The venetian system has the same depth as the spherical tracker so that there are 8 horizontal slats covering the window for that system. The goal of the choice of geometry was to easily assess the impact of adding a degree of freedom to the system on the energy and lighting performance. 

While the authors recognize that the size of the venetian shading system does not correspond to the most common size of venetian blinds (~8 cm deep slats is common for external systems), the authors would like to dispute the fact that the chosen size is not found in practice. Buildings such as the recently inaugurated French Institute of Science and Technology for Transport, Development and Networks (IFFSTAR) building in Champs sur Marne, France present horizontal slats for shading of a size similar to the ones implemented in the study (see https://fr.wikipedia.org/wiki/Fichier:Batiment-bienvenue-4397-020_2008.jpg). Other such examples are available in practice and in literature. Therefore, the authors consider that the system can be characterized as “commercially available”. 

10. Lines 210-212: the authors might consider modifying the text in this part of the manuscript to indicate that the awning is the only system that allows to have a completely unobstructed view to the outside. The other two systems allow to totally close the window, but the current text might confuse the reader as it seems that only the awning allows it.

The two sentences on lines 210-212 were modified to clarify the window obstruction according to the reviewer’s comment:

“The awning is the only system that is able to provide a completely unobstructed view to the outside. The venetian shades and the spherical tracker system always remain in front of the glazing even in the fully open position.”

11. Line 234: please correct the error in the subtitle of the text (now in line with the previous paragraph).

The paragraph number was corrected to “2.3.1 thermal energy” 

12. Lines 239-240: please indicate the heating and cooling temperature set-backs.

The setbacks have been precised in the following sentence: “The heating set-back is set at 15°C while the cooling set-back is set at 32°C.”

13. Lines 241-244: If the solar radiation calculation does not consider the daylight simulations (hence, it is guessed also the position of the shading devices to guarantee the minimum and maximum illuminance levels at the workplane), do energy results include the dynamic positions of the shading systems? If not, how can energy results be different? If yes, how is the information related to the shading device integrated into the energy simulations? It is suggested to the authors to better explain this process to avoid confusing the readers.

The electric energy necessary to maintain the thermal conditions (20C minimum for heating and 26C maximum for cooling) is directly linked to the dynamic positioning of the shades. In fact, for each sun hour, the energy necessary to maintain these thermal conditions is calculated for all the possible dynamic positions of the shading systems. The results of the energy simulation therefore change based on how open or closed the shades are. In practice, this means that for the spherical tracking shades, there are 81 thermal independent energy simulations performed at each sun hour. 

The solar radiation study does not consider daylight in the sense that energyplus does not perform the daylighting calculations. The daylighting is done using Radiance. For each sun hour, the illuminance on the work plane is calculated for each position of the shades. Therefore, both the energy simulations and daylighting simulation are directly related to the position of the shading system since the methodology gives the value of energy consumption and daylighting for each position of the shades. At each sun hour, the results for energy and daylight are interpolated (see figure 4) so that the optimization process can be used to decide what is the best position of the shading system to minimize the energy consumption and maintain the quantity of daylight set in the constraints.

The authors have clarified the presentation of the methodology following the comment #1 of the reviewer on the organization of the manuscript. 

14. Lines 254-255: please correct the units.

The S.I. unit for the R-value is K·m²/W. The authors are unsure about what the comment of the reviewer applies to. The units have been modified to appear in the same format “K·m²/W”. They were formatted slightly differently in the initial submission. 

15. Line 266: the use of daylight illuminance as a metric for measure daylight quality is not appropriate. Daylight quality also refers to distribution, color, dynamics. Please refer only to daylight quantity.

The term quality was replaced with quantity at the line 266 and at three other instances of the paper where it was used in a similar way. The three instances are in

- Introduction

- Optimization system for control of shades

- Selected energy and daylighting control variables

16. Line 266: if there is a sensor for each square in figure 1, what are the four sensors indicated in the figure?

The four sensors indicated in Figure 1 are the electric lighting sensors. They are used to adjust the lighting level when the daylight does not allow to reach the 500 lux constraint, see Line 296-298 of initial submission for details. The caption of Figure 1 was changed to make this clearer to the following:

“Figure 1 (a) Perimeter office space – window (ww) in blue, work plane (wp) in yellow - exterior / interior ground (grd), walls (wll) and ceiling indicated – (b) The grid represents the daylighting analysis grid. There is one daylight sensor per square (green). The four yellow sensors are specific to artificial lighting and are used to determine when electric lighting is needed”

17. Lines 272-273: why is the information about window transmittance repeated?

The first mention of the window’s transmittance was deleted from the text. 

18. Lines 296-297: why dividing the space in 90x90 cm grid if only 4 sensors are used? What is the use of the remaining “squares”?

The information presented was not clear and the authors would like to thank the reviewer for pointing this out. There are two types of sensors on the grid: the daylighting sensors and the electric lighting sensors. there are 25 daylighting sensors (one per square) and only 4 electric lighting sensors. Figure 1 was modified to better reflect this. 

19. Line 302: from the description of the text it is argued that the artificial light is always on despite daylight provides the minimum illuminance on the desk plane? Is this correct? If yes, why not turning off the electric light?

Thank you for making us double check. The dimmer setting chosen in DIVA is so that when positioned at the lowest position of electric lighting the system still uses 1% of its max power, not 10%. The use of a dimmer was selected to provide the system with a variable level of electric lighting. The 1% of power consumption represents a standby power for the lighting system.

20. Lines 304-306: what do the authors mean with this sentence? That daylight diminishes with the depth of the room? Please clarify. In addition, why two lighting systems (e.g., front and back of the room) were not modeled? This would have saved electric light energy as occupants close to the window would have probably had the necessary illuminance level from daylight alone.

Yes, the authors mean that the daylight diminishes with the depth of the room. The sentence was clarified so that future reader will understand easier what is meant. It now reads: “Given the geometry of the room and the presence of a large window on one wall only, daylight diminishes with the depth of the room.”

The authors agree that in an ideal situation, it would make sense to simulate two independent electric lighting systems. However, the constraint of the optimization is to produce a 500 lux average work plane and therefore a single system was used to compensate the lack of daylight. 

21. Lines 307-311: the explanation provided is somehow confusing. Are the authors saying that only energy demand for electric light is calculated, but not the final illuminance (natural + artificial) on the work plane? Please clarify the text.

The text was reworked to make it clearer for the reader how the system works. The electric lighting energy was calculated with the goal to meet the average lighting constraint on the work-plane. The assumption is that the electric lighting provides uniform lighting over the work plane, i.e. it can raise the average work-plane illuminance to 500 lux if the daylight is not enough to reach this level. The paragraph now reads: 

“The algorithm for electric lighting is open loop; it does not feed back into the daylighting assessment. The electric lighting energy was calculated with the goal to meet the average lighting constraint on the work-plane. If the average work plane illuminance provided by natural daylight is below the 500 lux required by the constraint, the electric lighting compensates with the appropriate amount of lighting to reach the target 500 lux. The electric lighting is provided uniformly to the work plane so the average illuminance will increase to 500 lux. It is not a multi-zone lighting and cannot compensate for the diminution of daylight with the depth of the room. That more advanced feature should be integrated in further lighting specific studies. Since the simulation is open loop, the algorithm does not iterate to evaluate the final work plane illuminance. This final lighting distribution would be found by iterating over the sum of the daylight and electric lighting contributions. The current algorithm guarantees that the average illuminance on the work plane of this new lighting (electric and daylight) is 500 lux. The occupancy schedule is set to the same hours as the thermal schedule (Figure 3), with the exception that it considers daylight-savings time.” 

22. Line 412: glare is included into the visual comfort.

The word glare was removed.

23. Section 2.5-2.6: It is not clear how the optimization process works: considering the two constrains (visual and thermal comfort), which is prioritized? Is the information related to visual comfort (more light and more solar radiation) integrated into the thermal simulation?

Two aspects of comfort are evaluated: thermal and visual. They are taken into consideration in different ways due to the specificity of their respective simulation methods.

1. Thermal comfort is considered by the setpoints of the thermal analysis performed in EnergyPlus. The heating temperature setpoint is set at 20C, which means the temperature of the room will not go below 20C when the weather is cold. The cooling temperature setpoint is set at 26C which means the temperature will not go above 26C when the weather is warm. Those two temperature setpoints guarantee the thermal comfort of the occupants. No matter what the shading typology is, no matter if it’s the baseline case, the thermal comfort is maintained by the setpoints and the same temperature at any given moment is expected in the room. The shading systems will have an impact on the amount of energy necessary to maintain this setpoint temperature. Therefore, thermal comfort is implicitly guaranteed in this simulation. 

2. The daylight level is calculated explicitly from the positions of the shades since it is not a product of the EnergyPlus simulation. The constraint is therefore explicit in the optimization system. From the work plane illuminance info, two parameters are calculated: the average illuminance and the maximum illuminance. The former is an indicator of the level of light and the latter a proxy for glare. The visual comfort is coupled with the thermal analysis since each shading systems will have several possible actuation positions and since each position influences both the thermal load and the level of daylight transmitted to the room. 

The assumption with the thermal comfort is that the setpoints temperatures are by default respected. Therefore, thermal comfort is not the driver of the optimization. In other words, for any solar radiation or shading situation, the assumption is that enough energy inputted in the system can lead to air temperatures that respect the setpoints. However, not all of these situations will lead to satisfying visual comfort. Therefore, visual comfort is the controlling constraint in the optimization. 

This clarification was added to sections 2.5 and 2.6 that explain how both components of comfort are taken into account.

24. Line 419: how is the validation performed? Please explain it in detail for reproducibility.

The term validation might be misused here. The breakdown sequence of the methodology has been rewritten to reflect the decision-making process that happens at the final stage of the methodology. The final step of the methodology is a test of whether the results obtained by a shading system are satisfactory or not. In absolute terms, there is no end to improving the design of a shading system so the methodology can be looped forever. In practice, the design of a shading system should be stopped when the result satisfies the objectives of the designer.

25. Lines 441-442: in reality, the energy consumed by the three shading systems in the east orientation is very similar to the other two orientations. The decrease of energy consumption results smaller as the energy consumption of the base case on the east orientation is already lower compared to that of the same case in other orientations.

Thank you for making the precision point. The text was modified to not mislead the reader on the reasons for the 10% less reduction and to integrate the reasoning provided by the reviewer. It now reads: 

”The overall reduction of cooling demand of the shading systems on the east is about 10 percent lower than for the other orientations (Table 3). The energy consumed by the building system in the three shading cases in the east is similar to the two other orientations. The energy consumption of the baseline case is smaller on the east than for the other orientations, which leads to a decrease of energy consumption by the shading systems that is smaller than for the other two orientations.”

26. Line 444: the figure appears to be in kWh and not in MJ.

Thank you for pointing this out. The caption was modified to reflect the correct unit of kWh. 

27. Figure 6: it is surprising to see that the annual energy demand for heating for all the orientations and shading systems is very low at the latitude of Princeton. Can the authors explain this result? Was the room heated by the incoming solar radiation? If yes, was the solar radiation changed according to the operation of the shading devices?

The results indeed show a very low energy demand for heating. The hypothesis of the authors relates to the surface of the window and the existence of overheating due to solar radiations. As pointed in the description of the room and material properties the window to wall ratio of the design room is 65%. This value is on the higher end of what is found in practice, which indicates that solar radiations will play a significant role in heating the test room. In addition, the orientations included in this study (east, south-east and south) receive solar radiations every day for a large portion of the day (no northern orientation included in the study). Secondly, as presented on Line 253 to 255 of the initial submission, the R-value for the window is 13.3 K·m²/W. This value is the default of the thermal analysis software DIVA used in the study. In buildings, windows are usually the weak link of thermal insulation, in this study that is not the case. These factors explain the low value of the heating demand over the year. The baseline case that receives a lot of solar radiation has the lowest amount of heating demand and the highest demand for cooling. This confirm the hypothesis that the prevention of overheating due to solar radiation is the controlling mode of operation of the test room. The authors have included a note in the result section to highlight the low value sand a note in the discussion section to explain to readers the reasons for the low heating energy demand.

Results section:

 “Overall, the annual energy demand for heating for all the orientation and shading system is very low compared to the cooling energy demand.”

Discussion section: 

“For all cases and all orientations, the annual energy demand presents a very low energy demand for heating. Several factors explain this. With a large value of 65% of WWR, this result indicates that solar radiations play a significant role in heating the test room. In addition, the orientations included in this study (east, south-east and south) receive solar radiations every day for a large portion of the day (see Figure 7). Third, the R-value for the window (13.3 K·m²/W, default value in DIVA) is larger than average. In buildings, windows are usually the weak link of thermal insulation, in this study that is not the case. The baseline case receives a lot of solar radiation: it has the lowest amount of heating demand and the highest demand for cooling. This confirm that the prevention of overheating due to solar radiation is the controlling mode of operation of the test room.”

28. Figure 7: the total energy demand is characterized by the cooling needs. However, is it surprising to see that for the awning and the east orientation model, the higher energy demand occurs in the afternoon. How can this result be explained? By the thermal accumulation of indoor surfaces?

In the optimization framework proposed, the shades do not only reduce energy consumption, but they operate to maintain a fixed daylight quantity. The phenomenon pointed by the reviewer is a direct consequence of this daylight constraint. In figure 7, the baseline case sees a high energy input throughout the day on the east. However, the levels of daylight on the east are high only in the earliest hours (as shown in figure 8 and 9). The awning shades are the most efficient of the three shading systems at meeting the daylight constraint. It means that in the afternoon they are fully open or largely to let all the daylight into the room. As a result, the energy demand found in the baseline is also found in the awning shades. 

29. Figure 8: please add illuminance values to the colored legend to make the graph easier to understand.

In the legend of the graph the values 0, 500, 1000, 1500 and 2000 lux have been clearly labeled for clarity of understanding of the graph. Thank you for helping improve this figure.

30. Line 502: are the values displayed only related to daylight? What about the fact that the electric light was always on?

The daylight is controlled by the shading and variable. The electric lighting is used to adjust the quantity of light when natural daylight is insufficient. To clarify, the electric lighting consumes standby power during the hours of operation of the building. As mentioned in a previous comment, this is the reason the lighting appears to be “always on”.

31. Line 511: reference to figure 7 is wrong.

The reference to the figure was corrected. 

32. Lines 640-642: can the authors expand on this topic? Wouldn’t the awning be the best option in terms of quantity of view out as it is the only one fully openable?

The reviewers are correct. Thank you for noticing this imprecise statement. The spherical tracker is better than the venetian shades but overall the awning shades with their ability to provide unobstructed views are the better option. The text of this paragraph was modified to clarify and better convey to the reader the finding of the study and the benefits of the new methodology presented. 

“The choice of comfort criteria to be used as input in the optimization system influences the outcome of the analysis. A choice of a lower average work-plane illuminance constraint may have produced different results, for instance. The venetian shadings and the spherical tracking shades produce average illuminances consistently in the interval 220 lx-330 lx in our study. Setting the constraint within that range could have shown one of these two shading systems as the best overall system. However, this would have certainly obstructed the fact that those are less versatile than the awning system in their tested configurations. They provide low average daylight quantities because they do not allow unobstructed views to the outside and therefore are limited in the amount of daylight they can let in diffuse light situations. Similarly, the spherical tracking shades are intuitively better than venetian shades for unobstructed views from the inside to the outside of the space, due to their ability to twist out of plane. Overall however, the awning shades are the best performer of the three types of shades due to their ability to be fully openable. The present methodology can help refine the design of this shading system. In their present configuration, the awning shades will most likely need a lot of maintenance due to being designed as a fabric dynamic system. This system is very susceptible to wind and rain, which make it unrealistic to practical implementation. Finally, their energy performance can be improved to match what the other two systems provide.”

33. Line 652: it is suggested to use the wording “and maintain the visual comfort of user into acceptable ranges”..

The text was modified according to the suggestion of the reviewer and now reads:

“This study introduces a methodology for the evaluation and optimization of the performance of shading systems to reduce heating, cooling, and lighting energy demands, and maintain the visual comfort of user into acceptable ranges.”

34. Table 9: it is not clear how the percentages are calculated. Please explain in the text or in the caption of the table.

Table 9 presents the information of figure 10 in a table form and makes the average of the results. The title of the table was changed to “Frequency of average work plane daylight quantity matching the 500 lx constraint (% of annual sun hours)” and the caption has an additional sentence to clarify how the values were calculated: “The values represent the frequency over the 4306 annual sun hours that the average work plane illuminance will be in the 475-525 lux range.”

35. Conclusions: could the authors better explain how researchers and practitioners can apply the suggested methodology in practice? In a real building, how would the suggested control algorithm work?

The authors have clarified the purpose of the methodology earlier on in the paper. However, while the focus of the paper is more precise, the authors have added the paragraph below in the conclusions section to specifically address the point raised by the reviewer. 

“The implementation for the control of existing dynamic shade is possible with this methodology. Once the methodology has been implemented to refine the design of shading system, this new solution must be operated in the actual building. The presented methodology is not a control algorithm for dynamic shading, but it provides a path for the development of more capable shading systems. However, the methodology can be adapted to be an operation-oriented algorithm. This change to the methodology can be carried in a two-step process. For a given building and room, the interpolated function should be trained for all the possible weather types (e.g. clear, mixed or overcast sky). The optimization should then be run at each time step by selecting the instantaneous weather condition. This would therefore require that sensors determine the weather condition at the current time and that a computing unit ran the optimization system with the correct set of interpolated functions. The issue with this operation method is that it requires knowledge of the building’s floor plan and have access to geometric models compatible with EnergyPlus and Radiance. Future work should focus on circumventing this limitation of the operation while still be able to access the high multi-dimensional capabilities that using this design methodology provides.”

 

Manuscript PONE-D-19-30597

Answer to Reviewer #2

In this document, the authors address the comments of reviewer 2. The authors would like to thank reviewer 2 for his/her comments aimed at improving the quality of the manuscript. 

General comment:

Reviewer #2: The paper investigated the Genetic Algorithm to optimize external dynamic shading in the USA. The study is rich in content and explanation. Generally, the manuscript is well written and the research novelty is linked to the development of dynamic optimization method. However, there are some comments should be considered to improve the research quality:

Thank you for your encouraging comment. The authors would like to mention that they have proceeded to a reorganization of the paper to better highlight the methodology section independently from the case study. In addition, the aim of the paper was slightly adjusted to reflect the main contribution of the manuscript which is the introduction of a methodology to compare and contrast the performance of different dynamic shading typologies. This change is reflected in the title of the paper that now reads: “Iterative optimization framework to advance the design of new dynamic shading typologies” as well as in the additional section 4.3 that presents results on the usefulness of each degree of freedom for the overall performance of their associated shading systems.

Detailed comments:

1- The abstract mentioned

"The heating, cooling, artificial lighting and daylight (work plane illuminance) performances are evaluated with Radiance and EnergyPlus based on local weather data."

However, the results mainly focus on daylight analysis while heating, cooling, artificial lighting are superficially covered. A proper balance among different factors should be provided.

The abstract was re-written to integrate the revised focus of the article and the comment of the reviewer. The abstract now reads: 

“Dynamic solar shading has the potential to dramatically reduce the energy consumption in buildings while at the same time providing an improved access to natural daylight for its occupants. Many new typologies of shading systems that have appeared recently, but it is difficult to compare those new systems to existing typologies due to control algorithm being rule-based as opposed to performance driven. Since solar shading is a design problem, there is no single right answer. What is the metric to determine if a system has reached its optimal kinematic design? Shading solution should come from a thorough iterative and comparative process. This paper provides an original and flexible framework for the design and performance optimization of dynamic shading systems based on interpolation and genetic algorithm global minimization. The methodology departs from existing rule-based strategies and applies to existing and to complex shading systems with multiple degree-of-freedom mobility. The strategy for control is centered on meeting comfort targets for work plane illuminance while minimizing the energy needed to operate space. The electric demand for thermal comfort and work plane daylight quantity (illuminance) performances are evaluated with Radiance and EnergyPlus based on local weather data. Applied to a case study of three typologies of dynamic shading, the results of the methodology inform the usefulness and quality of each degree-of-freedom of the kinematic system. The case study exemplifies the iterative benefits of the methodology by providing detailed analytics on the behavior of the shades. Designers of shading systems can use this framework to evaluate their design and compare them to existing shading systems. This allows creativity to be guided so that the building occupants benefits from the innovation in the field.”

2- The paper should improve its literature with the latest research to support the research decisions, for example:

The study mentioned that the set point of the lighting system is 500 lx without providing an explanation. Also, the study mentioned "The simulated perimeter office is 5 m deep, 4.5 m wide, and 3.2 m high" without providing support from the literature. Including the study below would help to support this claim.

• Luo, Y., Zhang, L., Su, X., Liu, Z., Lian, J., & Luo, Y. (2019). Improved thermal-electrical-optical model and performance assessment of a PV-blind embedded glazing façade system with complex shading effects. Applied Energy, 255, 113896.

• Al-Obaidi, K. M., Munaaim, M. A. C., Ismail, M. A., & Rahman, A. M. A. (2017). Designing an integrated daylighting system for deep-plan spaces in Malaysian low-rise buildings. Solar Energy, 149, 85-101.

Also, the study mentioned about, the algorithm for electric lighting is open-loop; it does not feedback into the daylighting assessment. However, the study does not discuss different feedback systems as open-loop or closed-loop in dynamic shading systems. Including the study below would help to support this claim.

• Al-Masrani, S. M., & Al-Obaidi, K. M. (2019). Dynamic shading systems: A review of design parameters, platforms and evaluation strategies. Automation in construction, 102, 195-216.

• Ayoub, M. (2018). Integrating illuminance and energy evaluations of cellular automata controlled dynamic shading system using new hourly-based metrics. Solar Energy, 170, 336-351.

The authors would like to thank the reviewer for providing the appropriate literature to support the choice of the room size. Those references were included in section 3.1. Similarly, the authors have included the following sentence in Section 2.2 “In that sense, it is similar to the open loop operation system described in [13].” To integrate the comment about the open loop algorithm.

3- The study needs to explain the outdoor daylight and sky conditions in Mercer County, New Jersey, USA that would control the study outcomes.

The authors have added a figure (Figure 3) to explain the outdoor daylight and sky conditions in mercer county throughout the year. Thank you for pointing out this opportunity to drive the point that the reduction of environmental daylight for human comfort is substantial. 

4- The study mentioned about 4 sensors as shown in Figure 1 but I couldn't see the results of each sensor.

The four sensors indicated in Figure 1 are the electric lighting sensors. They are used to adjust the lighting level when the daylight does not allow to reach the 500 lux constraint, see Line 296-298 of initial submission for details. The constraint of the optimization is to produce a 500 lux average work plane and therefore a single system was used to compensate the lack of daylight. The daylight quantity produced by the electric lighting system is given as the difference between the average daylight quantity provided and the objective of 500 lux of work plane illuminance. This value can be assessed by looking at figure 9 that represents the average workplane daylight quantity for the three shading systems and for the three orientations of the test room. 

5- The study mentioned ""The total run-time of the analysis was 73 minutes on an Intel i7 4 core CPU at 4.20GHz. This total time is decomposed in 21 minutes for the daylighting analysis in Radiance, 42 minutes for the thermal analysis in EnergyPlus and 10 minutes for the optimization in MATLAB."

How could this statement help the outcomes of this research?

The authors believe this information provides the reader unfamiliar with one of the aspects of the study with a benchmark of what each section of the methodology requires in terms of computation. In addition, highlighting the ratio between optimization and simulation allows to point out that the methodology is composed of two parts, simulation and optimization. Once the simulation is performed, the optimization can be run independently for varying values of the constraints chosen by the designer. The following sentence was added in section 5.1 to make this point clearer for the reader:

“For a given case study, once the simulations have been performed, the optimization can be run for any value of the illuminance constraint without having to perform the simulations again”

6- Figure 2, model C, it is unclear to understand the model movement and rotation.

The authors have improved the figure to make it more understandable. Namely, the azimuth degree of freedom was clarified and both pictures were made larger.

---

## [Editor Report · Decision Letter 1]

20 Feb 2020

PONE-D-19-30597R1

Occupant-centered optimization framework to evaluate and design new dynamic shading typologies

PLOS ONE

Dear Dr. Charpentier,

following your email, please re-submit your manuscript including Figure 3.

Best regards

Marco Lepidi

Marco Lepidi, PhD

Associate Professor, Academic Editor of PLOS ONE

Dipartimento di Ingegneria Civile, Chimica ed Ambientale

Università degli Studi di Genova (Italy)

Mail: Marco.Lepidi@unige.it

Website: http://www3.dicca.unige.it/mlepidi/index.html

Google Scholar: http://scholar.google.it/citations?user=uePAdVUAAAAJ&hl=it

---

## [Author Response · Author response to Decision Letter 1]

24 Feb 2020

Manuscript PONE-D-19-30597

Answer to Reviewer #1

In this document, the authors address the comments of reviewer 1. The authors would like to thank reviewer 1 for his/her thorough and very helpful comments aimed at improving the quality of the manuscript. We are grateful for the time and effort spend by reviewer 1. We hope the changes we have made to the manuscript will be satisfactory as we believe the resulting manuscript is much improved.

General comment:

Reviewer #1: Investigations on the operation of dynamic shading devices are important nowadays considering the more and more widespread use of such technology in buildings. The authors describe a methodology for the control of dynamic shading devices to minimize energy use while guaranteeing the visual comfort of occupants, as well as its application for the evaluation of three shading systems for thee orientations (east, south east and south).

Detailed comments:

1.General comment: considering that the focus of the paper is primarily on the description of a methodology for the evaluation of the performance of dynamic shading devices, it is suggested to revise the organization of the manuscript and of its sections. For example, the methodology could be described first, while its application to the selected shading options later. Similarly, it is suggested to change the title to better highlight the scope of the paper. The wording “non-deterministic” present in the title is not reported in the text of the manuscript and therefore is not properly explained.

The authors would like to thank the reviewer for the challenging comment. As a result, two major changes have been brought to the paper. 

First the title of the manuscript was changed to “Occupant-centered optimization framework to evaluate and design new dynamic shading typologies”. The authors think it now better reflects the contribution of the paper which is to evaluate and compare the performance of dynamic shading systems.

Second, the manuscript was reorganized as suggested by the reviewer. The initial methodology section that contained both the detail of the case study and the detail of the methodology was split in two sections. The presentation of the methodology comes first in section 2: “Methodology to evaluate and compare dynamic shading typologies”. The detail of the case study comes second in section 3: “detail of the case study”. The result of this modification is that the reader can now clearly understand the methodology independent of the details of the case study. 

2. General comment: The comparison with a baseline is appreciated and required to understand the value of the proposed solution. However, it would be more useful if the baseline was not an unshaded window (despite it is recognized its widespread presence in buildings in the USA), but the same types of blinds with a different and “more common” control algorithm, considering that the control algorithm suggested is the novelty of this paper. It is suggested to expand the comparison to a different (or additional) baseline model.

The authors agree with the comment of the reviewer that in the case of the presentation of a new control algorithm a suitable baseline would be an algorithm used in practice. However, the paper is aimed at being a comparison between different typologies of dynamic shading systems. By re-organizing the manuscript, the authors additionally made the focus of the paper more precise. The paper now focuses on comparing the performances of shading systems to enable increased creativity in their design. An additional section (4.3 use of each degree of freedom) was added to make this refocus clearer to the reader. The new section provides a new way to look at the design of a shading system by allowing to visualize the use of each degree-of-freedom of the shading typologies. A high usage, i.e. frequent motion, is correlated to a highly efficient shading system while a low usage translates an issue with the design of the shading system.

The unshaded window is used as a baseline in this case because it provides a representation of the most common scenario for daylighting. In this refocused paper the goal is not to provide a control algorithm but to substantiate the usefulness of the combination of higher expectations and advanced tools for the design of dynamic shading typologies. 

3. Abstract, line 39: it is surprising to read that electricity is used to maintain the constraint about daylight level. It is assumed that the constraint is about light level and not daylight level, considering that daylight is not linked to energy use.

Thank you for pointing out this imprecision. The overall lighting level is adjusted by the electric lighting, the daylight level is fixed by environmental conditions. 

4. Abstract: why only results associated to the spherical solar tracking are presented in the abstract? Is the primary goal of the paper to evaluate a specific shading device? It is suggested to reformulate the abstract to be focused on the actual goal of the paper.

The goal of the paper is to present the methodology and to display the results of its implementation for standard and novel dynamic shading systems. The reviewer is right in pointing the inconsistency of the abstract. The last part of the abstract was therefore reformulated to reflect the broader goal of the paper. It previously was: 

“When tested on a prototype design of spherical tracking dynamic external shades, our methodology led to a 62% decrease in annual cooling energy demand, but an 81% increase in lighting energy to maintain the constraint daylight levels. Overall this shading system reduced the annual energy demand by 45% on average over the three orientations tested.”

And now reads as: 

“Over the three shading typologies tested, the average energy consumption is reduced by 44%. The tradeoff appears with a decrease of average annual cooling of 60% associated with an average increase in electric lighting of 77%. The methodology highlights the awning shades presented in the paper as the best overall performer with a good ability to control natural daylight and an overall reduction of 33% of the annual energy consumption over the three orientations tested.”

5. Lines 63-64: the current wording of this sentence is not very clear. What did the authors mean with it?

Thank you for pointing this sentence’s lack of clarity. The sentence was meant to summarize the significance of using a holistic approach to solar shading that would revolve around the improvement of the overall comfort of occupants. Psychological aspects are key to individual comfort and the proposed methodology with its flexibility of constraints can be used to integrate these notions into a formal framework. The sentence was modified and now reads: “The parameters of individual comfort are interdependent due to psychological factors. There is therefore great significance in using a holistic approach to solar shading revolving around the improvement of the overall comfort of occupants”. 

An additional sentence referring to psychological ties and the advantages of the methodology was added in the introduction: “In addition, since psychological aspects are key to individual comfort, the proposed methodology can be improved to integrate new metrics of performance.

6. Line 136: remove electric lighting from the sentence as it does not affect the thermal comfort (unless radiation from operating electric light is included into the modeling?).

The aspects of comfort under study in this paper are thermal and lighting comfort. The energy in question here is the overall consumption of the building system for to maintain a global occupant comfort level. The sentence was therefore modified to make this clearer:

“Energy demand for heating, cooling and electric lighting are calculated to maintain a fixed level of occupant comfort (thermal and lighting).” 

7. Lines 162-163: this information has already been presented. Please describe the simulation model in one section of the manuscript only (it is suggested to dedicate to it a specific subsection).

Thank you for pointing this redundancy. The first mention of the room size was removed in the introduction to section 2 detailing the content of the section.

8. Lines 173-177: revise the caption of the figure.

The caption was revised and now reads:

“Figure 1 (a) Perimeter office space – window (ww) in blue, work plane (wp) in yellow - exterior / interior ground (grd), walls (wll) and ceiling indicated – (b) daylighting grid with a daylight sensor per square, four artificial lighting sensors determine when lighting is needed”

9. Line 189: the authors indicate to have modeled three blind solutions that are similar to commercially available systems. However, especially for the modeling of the venetian blinds, it appears that too many simplifications were done. In the commercially available system indicated by the authors, for example, the slats are not flat but present more elaborated geometries. In addition, a depth of 27.5 cm (line 206) is considered too big compared to commercially available systems. Why did the authors model the slats in this way? Why not choosing less deep slats and increasing the number of slats to cover the height of the window? Also, the dimensions of the spherical solar tracker components are pretty big. In practice, this could cause problems with the mechanical systems of the actuators.

The size of the spherical tracker corresponds to the prototyped size of the novel shading system currently being developed by the authors. The system is based on active bending of shell structures and is actuated using shape memory alloy wires. In the authors’ prototype, the modules are between 25 and 30 cm tall and presented on the façade as shown in figure 2 of the initial submission. The choice of 27.5 cm corresponds to a division by 8 of the vertical height of the window. The initial manuscript mentions the relationship with the ongoing work of the authors at L.200-201. 

The size of the spherical tracker and of the venetian shades are related. The venetian system has the same depth as the spherical tracker so that there are 8 horizontal slats covering the window for that system. The goal of the choice of geometry was to easily assess the impact of adding a degree of freedom to the system on the energy and lighting performance. 

While the authors recognize that the size of the venetian shading system does not correspond to the most common size of venetian blinds (~8 cm deep slats is common for external systems), the authors would like to dispute the fact that the chosen size is not found in practice. Buildings such as the recently inaugurated French Institute of Science and Technology for Transport, Development and Networks (IFFSTAR) building in Champs sur Marne, France present horizontal slats for shading of a size similar to the ones implemented in the study (see https://fr.wikipedia.org/wiki/Fichier:Batiment-bienvenue-4397-020_2008.jpg). Other such examples are available in practice and in literature. Therefore, the authors consider that the system can be characterized as “commercially available”. 

10. Lines 210-212: the authors might consider modifying the text in this part of the manuscript to indicate that the awning is the only system that allows to have a completely unobstructed view to the outside. The other two systems allow to totally close the window, but the current text might confuse the reader as it seems that only the awning allows it.

The two sentences on lines 210-212 were modified to clarify the window obstruction according to the reviewer’s comment:

“The awning is the only system that is able to provide a completely unobstructed view to the outside. The venetian shades and the spherical tracker system always remain in front of the glazing even in the fully open position.”

11. Line 234: please correct the error in the subtitle of the text (now in line with the previous paragraph).

The paragraph number was corrected to “2.3.1 thermal energy” 

12. Lines 239-240: please indicate the heating and cooling temperature set-backs.

The setbacks have been precised in the following sentence: “The heating set-back is set at 15°C while the cooling set-back is set at 32°C.”

13. Lines 241-244: If the solar radiation calculation does not consider the daylight simulations (hence, it is guessed also the position of the shading devices to guarantee the minimum and maximum illuminance levels at the workplane), do energy results include the dynamic positions of the shading systems? If not, how can energy results be different? If yes, how is the information related to the shading device integrated into the energy simulations? It is suggested to the authors to better explain this process to avoid confusing the readers.

The electric energy necessary to maintain the thermal conditions (20C minimum for heating and 26C maximum for cooling) is directly linked to the dynamic positioning of the shades. In fact, for each sun hour, the energy necessary to maintain these thermal conditions is calculated for all the possible dynamic positions of the shading systems. The results of the energy simulation therefore change based on how open or closed the shades are. In practice, this means that for the spherical tracking shades, there are 81 thermal independent energy simulations performed at each sun hour. 

The solar radiation study does not consider daylight in the sense that energyplus does not perform the daylighting calculations. The daylighting is done using Radiance. For each sun hour, the illuminance on the work plane is calculated for each position of the shades. Therefore, both the energy simulations and daylighting simulation are directly related to the position of the shading system since the methodology gives the value of energy consumption and daylighting for each position of the shades. At each sun hour, the results for energy and daylight are interpolated (see figure 4) so that the optimization process can be used to decide what is the best position of the shading system to minimize the energy consumption and maintain the quantity of daylight set in the constraints.

The authors have clarified the presentation of the methodology following the comment #1 of the reviewer on the organization of the manuscript. 

14. Lines 254-255: please correct the units.

The S.I. unit for the R-value is K·m²/W. The authors are unsure about what the comment of the reviewer applies to. The units have been modified to appear in the same format “K·m²/W”. They were formatted slightly differently in the initial submission. 

15. Line 266: the use of daylight illuminance as a metric for measure daylight quality is not appropriate. Daylight quality also refers to distribution, color, dynamics. Please refer only to daylight quantity.

The term quality was replaced with quantity at the line 266 and at three other instances of the paper where it was used in a similar way. The three instances are in

- Introduction

- Optimization system for control of shades

- Selected energy and daylighting control variables

16. Line 266: if there is a sensor for each square in figure 1, what are the four sensors indicated in the figure?

The four sensors indicated in Figure 1 are the electric lighting sensors. They are used to adjust the lighting level when the daylight does not allow to reach the 500 lux constraint, see Line 296-298 of initial submission for details. The caption of Figure 1 was changed to make this clearer to the following:

“Figure 1 (a) Perimeter office space – window (ww) in blue, work plane (wp) in yellow - exterior / interior ground (grd), walls (wll) and ceiling indicated – (b) The grid represents the daylighting analysis grid. There is one daylight sensor per square (green). The four yellow sensors are specific to artificial lighting and are used to determine when electric lighting is needed”

17. Lines 272-273: why is the information about window transmittance repeated?

The first mention of the window’s transmittance was deleted from the text. 

18. Lines 296-297: why dividing the space in 90x90 cm grid if only 4 sensors are used? What is the use of the remaining “squares”?

The information presented was not clear and the authors would like to thank the reviewer for pointing this out. There are two types of sensors on the grid: the daylighting sensors and the electric lighting sensors. there are 25 daylighting sensors (one per square) and only 4 electric lighting sensors. Figure 1 was modified to better reflect this. 

19. Line 302: from the description of the text it is argued that the artificial light is always on despite daylight provides the minimum illuminance on the desk plane? Is this correct? If yes, why not turning off the electric light?

Thank you for making us double check. The dimmer setting chosen in DIVA is so that when positioned at the lowest position of electric lighting the system still uses 1% of its max power, not 10%. The use of a dimmer was selected to provide the system with a variable level of electric lighting. The 1% of power consumption represents a standby power for the lighting system.

20. Lines 304-306: what do the authors mean with this sentence? That daylight diminishes with the depth of the room? Please clarify. In addition, why two lighting systems (e.g., front and back of the room) were not modeled? This would have saved electric light energy as occupants close to the window would have probably had the necessary illuminance level from daylight alone.

Yes, the authors mean that the daylight diminishes with the depth of the room. The sentence was clarified so that future reader will understand easier what is meant. It now reads: “Given the geometry of the room and the presence of a large window on one wall only, daylight diminishes with the depth of the room.”

The authors agree that in an ideal situation, it would make sense to simulate two independent electric lighting systems. However, the constraint of the optimization is to produce a 500 lux average work plane and therefore a single system was used to compensate the lack of daylight. 

21. Lines 307-311: the explanation provided is somehow confusing. Are the authors saying that only energy demand for electric light is calculated, but not the final illuminance (natural + artificial) on the work plane? Please clarify the text.

The text was reworked to make it clearer for the reader how the system works. The electric lighting energy was calculated with the goal to meet the average lighting constraint on the work-plane. The assumption is that the electric lighting provides uniform lighting over the work plane, i.e. it can raise the average work-plane illuminance to 500 lux if the daylight is not enough to reach this level. The paragraph now reads: 

“The algorithm for electric lighting is open loop; it does not feed back into the daylighting assessment. The electric lighting energy was calculated with the goal to meet the average lighting constraint on the work-plane. If the average work plane illuminance provided by natural daylight is below the 500 lux required by the constraint, the electric lighting compensates with the appropriate amount of lighting to reach the target 500 lux. The electric lighting is provided uniformly to the work plane so the average illuminance will increase to 500 lux. It is not a multi-zone lighting and cannot compensate for the diminution of daylight with the depth of the room. That more advanced feature should be integrated in further lighting specific studies. Since the simulation is open loop, the algorithm does not iterate to evaluate the final work plane illuminance. This final lighting distribution would be found by iterating over the sum of the daylight and electric lighting contributions. The current algorithm guarantees that the average illuminance on the work plane of this new lighting (electric and daylight) is 500 lux. The occupancy schedule is set to the same hours as the thermal schedule (Figure 3), with the exception that it considers daylight-savings time.” 

22. Line 412: glare is included into the visual comfort.

The word glare was removed.

23. Section 2.5-2.6: It is not clear how the optimization process works: considering the two constrains (visual and thermal comfort), which is prioritized? Is the information related to visual comfort (more light and more solar radiation) integrated into the thermal simulation?

Two aspects of comfort are evaluated: thermal and visual. They are taken into consideration in different ways due to the specificity of their respective simulation methods.

1. Thermal comfort is considered by the setpoints of the thermal analysis performed in EnergyPlus. The heating temperature setpoint is set at 20C, which means the temperature of the room will not go below 20C when the weather is cold. The cooling temperature setpoint is set at 26C which means the temperature will not go above 26C when the weather is warm. Those two temperature setpoints guarantee the thermal comfort of the occupants. No matter what the shading typology is, no matter if it’s the baseline case, the thermal comfort is maintained by the setpoints and the same temperature at any given moment is expected in the room. The shading systems will have an impact on the amount of energy necessary to maintain this setpoint temperature. Therefore, thermal comfort is implicitly guaranteed in this simulation. 

2. The daylight level is calculated explicitly from the positions of the shades since it is not a product of the EnergyPlus simulation. The constraint is therefore explicit in the optimization system. From the work plane illuminance info, two parameters are calculated: the average illuminance and the maximum illuminance. The former is an indicator of the level of light and the latter a proxy for glare. The visual comfort is coupled with the thermal analysis since each shading systems will have several possible actuation positions and since each position influences both the thermal load and the level of daylight transmitted to the room. 

The assumption with the thermal comfort is that the setpoints temperatures are by default respected. Therefore, thermal comfort is not the driver of the optimization. In other words, for any solar radiation or shading situation, the assumption is that enough energy inputted in the system can lead to air temperatures that respect the setpoints. However, not all of these situations will lead to satisfying visual comfort. Therefore, visual comfort is the controlling constraint in the optimization. 

This clarification was added to sections 2.5 and 2.6 that explain how both components of comfort are taken into account.

24. Line 419: how is the validation performed? Please explain it in detail for reproducibility.

The term validation might be misused here. The breakdown sequence of the methodology has been rewritten to reflect the decision-making process that happens at the final stage of the methodology. The final step of the methodology is a test of whether the results obtained by a shading system are satisfactory or not. In absolute terms, there is no end to improving the design of a shading system so the methodology can be looped forever. In practice, the design of a shading system should be stopped when the result satisfies the objectives of the designer.

25. Lines 441-442: in reality, the energy consumed by the three shading systems in the east orientation is very similar to the other two orientations. The decrease of energy consumption results smaller as the energy consumption of the base case on the east orientation is already lower compared to that of the same case in other orientations.

Thank you for making the precision point. The text was modified to not mislead the reader on the reasons for the 10% less reduction and to integrate the reasoning provided by the reviewer. It now reads: 

”The overall reduction of cooling demand of the shading systems on the east is about 10 percent lower than for the other orientations (Table 3). The energy consumed by the building system in the three shading cases in the east is similar to the two other orientations. The energy consumption of the baseline case is smaller on the east than for the other orientations, which leads to a decrease of energy consumption by the shading systems that is smaller than for the other two orientations.”

26. Line 444: the figure appears to be in kWh and not in MJ.

Thank you for pointing this out. The caption was modified to reflect the correct unit of kWh. 

27. Figure 6: it is surprising to see that the annual energy demand for heating for all the orientations and shading systems is very low at the latitude of Princeton. Can the authors explain this result? Was the room heated by the incoming solar radiation? If yes, was the solar radiation changed according to the operation of the shading devices?

The results indeed show a very low energy demand for heating. The hypothesis of the authors relates to the surface of the window and the existence of overheating due to solar radiations. As pointed in the description of the room and material properties the window to wall ratio of the design room is 65%. This value is on the higher end of what is found in practice, which indicates that solar radiations will play a significant role in heating the test room. In addition, the orientations included in this study (east, south-east and south) receive solar radiations every day for a large portion of the day (no northern orientation included in the study). Secondly, as presented on Line 253 to 255 of the initial submission, the R-value for the window is 13.3 K·m²/W. This value is the default of the thermal analysis software DIVA used in the study. In buildings, windows are usually the weak link of thermal insulation, in this study that is not the case. These factors explain the low value of the heating demand over the year. The baseline case that receives a lot of solar radiation has the lowest amount of heating demand and the highest demand for cooling. This confirm the hypothesis that the prevention of overheating due to solar radiation is the controlling mode of operation of the test room. The authors have included a note in the result section to highlight the low value sand a note in the discussion section to explain to readers the reasons for the low heating energy demand.

Results section:

 “Overall, the annual energy demand for heating for all the orientation and shading system is very low compared to the cooling energy demand.”

Discussion section: 

“For all cases and all orientations, the annual energy demand presents a very low energy demand for heating. Several factors explain this. With a large value of 65% of WWR, this result indicates that solar radiations play a significant role in heating the test room. In addition, the orientations included in this study (east, south-east and south) receive solar radiations every day for a large portion of the day (see Figure 7). Third, the R-value for the window (13.3 K·m²/W, default value in DIVA) is larger than average. In buildings, windows are usually the weak link of thermal insulation, in this study that is not the case. The baseline case receives a lot of solar radiation: it has the lowest amount of heating demand and the highest demand for cooling. This confirm that the prevention of overheating due to solar radiation is the controlling mode of operation of the test room.”

28. Figure 7: the total energy demand is characterized by the cooling needs. However, is it surprising to see that for the awning and the east orientation model, the higher energy demand occurs in the afternoon. How can this result be explained? By the thermal accumulation of indoor surfaces?

In the optimization framework proposed, the shades do not only reduce energy consumption, but they operate to maintain a fixed daylight quantity. The phenomenon pointed by the reviewer is a direct consequence of this daylight constraint. In figure 7, the baseline case sees a high energy input throughout the day on the east. However, the levels of daylight on the east are high only in the earliest hours (as shown in figure 8 and 9). The awning shades are the most efficient of the three shading systems at meeting the daylight constraint. It means that in the afternoon they are fully open or largely to let all the daylight into the room. As a result, the energy demand found in the baseline is also found in the awning shades. 

29. Figure 8: please add illuminance values to the colored legend to make the graph easier to understand.

In the legend of the graph the values 0, 500, 1000, 1500 and 2000 lux have been clearly labeled for clarity of understanding of the graph. Thank you for helping improve this figure.

30. Line 502: are the values displayed only related to daylight? What about the fact that the electric light was always on?

The daylight is controlled by the shading and variable. The electric lighting is used to adjust the quantity of light when natural daylight is insufficient. To clarify, the electric lighting consumes standby power during the hours of operation of the building. As mentioned in a previous comment, this is the reason the lighting appears to be “always on”.

31. Line 511: reference to figure 7 is wrong.

The reference to the figure was corrected. 

32. Lines 640-642: can the authors expand on this topic? Wouldn’t the awning be the best option in terms of quantity of view out as it is the only one fully openable?

The reviewers are correct. Thank you for noticing this imprecise statement. The spherical tracker is better than the venetian shades but overall the awning shades with their ability to provide unobstructed views are the better option. The text of this paragraph was modified to clarify and better convey to the reader the finding of the study and the benefits of the new methodology presented. 

“The choice of comfort criteria to be used as input in the optimization system influences the outcome of the analysis. A choice of a lower average work-plane illuminance constraint may have produced different results, for instance. The venetian shadings and the spherical tracking shades produce average illuminances consistently in the interval 220 lx-330 lx in our study. Setting the constraint within that range could have shown one of these two shading systems as the best overall system. However, this would have certainly obstructed the fact that those are less versatile than the awning system in their tested configurations. They provide low average daylight quantities because they do not allow unobstructed views to the outside and therefore are limited in the amount of daylight they can let in diffuse light situations. Similarly, the spherical tracking shades are intuitively better than venetian shades for unobstructed views from the inside to the outside of the space, due to their ability to twist out of plane. Overall however, the awning shades are the best performer of the three types of shades due to their ability to be fully openable. The present methodology can help refine the design of this shading system. In their present configuration, the awning shades will most likely need a lot of maintenance due to being designed as a fabric dynamic system. This system is very susceptible to wind and rain, which make it unrealistic to practical implementation. Finally, their energy performance can be improved to match what the other two systems provide.”

33. Line 652: it is suggested to use the wording “and maintain the visual comfort of user into acceptable ranges”..

The text was modified according to the suggestion of the reviewer and now reads:

“This study introduces a methodology for the evaluation and optimization of the performance of shading systems to reduce heating, cooling, and lighting energy demands, and maintain the visual comfort of user into acceptable ranges.”

34. Table 9: it is not clear how the percentages are calculated. Please explain in the text or in the caption of the table.

Table 9 presents the information of figure 10 in a table form and makes the average of the results. The title of the table was changed to “Frequency of average work plane daylight quantity matching the 500 lx constraint (% of annual sun hours)” and the caption has an additional sentence to clarify how the values were calculated: “The values represent the frequency over the 4306 annual sun hours that the average work plane illuminance will be in the 475-525 lux range.”

35. Conclusions: could the authors better explain how researchers and practitioners can apply the suggested methodology in practice? In a real building, how would the suggested control algorithm work?

The authors have clarified the purpose of the methodology earlier on in the paper. However, while the focus of the paper is more precise, the authors have added the paragraph below in the conclusions section to specifically address the point raised by the reviewer. 

“The implementation for the control of existing dynamic shade is possible with this methodology. Once the methodology has been implemented to refine the design of shading system, this new solution must be operated in the actual building. The presented methodology is not a control algorithm for dynamic shading, but it provides a path for the development of more capable shading systems. However, the methodology can be adapted to be an operation-oriented algorithm. This change to the methodology can be carried in a two-step process. For a given building and room, the interpolated function should be trained for all the possible weather types (e.g. clear, mixed or overcast sky). The optimization should then be run at each time step by selecting the instantaneous weather condition. This would therefore require that sensors determine the weather condition at the current time and that a computing unit ran the optimization system with the correct set of interpolated functions. The issue with this operation method is that it requires knowledge of the building’s floor plan and have access to geometric models compatible with EnergyPlus and Radiance. Future work should focus on circumventing this limitation of the operation while still be able to access the high multi-dimensional capabilities that using this design methodology provides.”

 

Manuscript PONE-D-19-30597

Answer to Reviewer #2

In this document, the authors address the comments of reviewer 2. The authors would like to thank reviewer 2 for his/her comments aimed at improving the quality of the manuscript. 

General comment:

Reviewer #2: The paper investigated the Genetic Algorithm to optimize external dynamic shading in the USA. The study is rich in content and explanation. Generally, the manuscript is well written and the research novelty is linked to the development of dynamic optimization method. However, there are some comments should be considered to improve the research quality:

Thank you for your encouraging comment. The authors would like to mention that they have proceeded to a reorganization of the paper to better highlight the methodology section independently from the case study. In addition, the aim of the paper was slightly adjusted to reflect the main contribution of the manuscript which is the introduction of a methodology to compare and contrast the performance of different dynamic shading typologies. This change is reflected in the title of the paper that now reads: “Iterative optimization framework to advance the design of new dynamic shading typologies” as well as in the additional section 4.3 that presents results on the usefulness of each degree of freedom for the overall performance of their associated shading systems.

Detailed comments:

1- The abstract mentioned

"The heating, cooling, artificial lighting and daylight (work plane illuminance) performances are evaluated with Radiance and EnergyPlus based on local weather data."

However, the results mainly focus on daylight analysis while heating, cooling, artificial lighting are superficially covered. A proper balance among different factors should be provided.

The abstract was re-written to integrate the revised focus of the article and the comment of the reviewer. The abstract now reads: 

“Dynamic solar shading has the potential to dramatically reduce the energy consumption in buildings while at the same time providing an improved access to natural daylight for its occupants. Many new typologies of shading systems that have appeared recently, but it is difficult to compare those new systems to existing typologies due to control algorithm being rule-based as opposed to performance driven. Since solar shading is a design problem, there is no single right answer. What is the metric to determine if a system has reached its optimal kinematic design? Shading solution should come from a thorough iterative and comparative process. This paper provides an original and flexible framework for the design and performance optimization of dynamic shading systems based on interpolation and genetic algorithm global minimization. The methodology departs from existing rule-based strategies and applies to existing and to complex shading systems with multiple degree-of-freedom mobility. The strategy for control is centered on meeting comfort targets for work plane illuminance while minimizing the energy needed to operate space. The electric demand for thermal comfort and work plane daylight quantity (illuminance) performances are evaluated with Radiance and EnergyPlus based on local weather data. Applied to a case study of three typologies of dynamic shading, the results of the methodology inform the usefulness and quality of each degree-of-freedom of the kinematic system. The case study exemplifies the iterative benefits of the methodology by providing detailed analytics on the behavior of the shades. Designers of shading systems can use this framework to evaluate their design and compare them to existing shading systems. This allows creativity to be guided so that the building occupants benefits from the innovation in the field.”

2- The paper should improve its literature with the latest research to support the research decisions, for example:

The study mentioned that the set point of the lighting system is 500 lx without providing an explanation. Also, the study mentioned "The simulated perimeter office is 5 m deep, 4.5 m wide, and 3.2 m high" without providing support from the literature. Including the study below would help to support this claim.

• Luo, Y., Zhang, L., Su, X., Liu, Z., Lian, J., & Luo, Y. (2019). Improved thermal-electrical-optical model and performance assessment of a PV-blind embedded glazing façade system with complex shading effects. Applied Energy, 255, 113896.

• Al-Obaidi, K. M., Munaaim, M. A. C., Ismail, M. A., & Rahman, A. M. A. (2017). Designing an integrated daylighting system for deep-plan spaces in Malaysian low-rise buildings. Solar Energy, 149, 85-101.

Also, the study mentioned about, the algorithm for electric lighting is open-loop; it does not feedback into the daylighting assessment. However, the study does not discuss different feedback systems as open-loop or closed-loop in dynamic shading systems. Including the study below would help to support this claim.

• Al-Masrani, S. M., & Al-Obaidi, K. M. (2019). Dynamic shading systems: A review of design parameters, platforms and evaluation strategies. Automation in construction, 102, 195-216.

• Ayoub, M. (2018). Integrating illuminance and energy evaluations of cellular automata controlled dynamic shading system using new hourly-based metrics. Solar Energy, 170, 336-351.

The authors would like to thank the reviewer for providing the appropriate literature to support the choice of the room size. Those references were included in section 3.1. Similarly, the authors have included the following sentence in Section 2.2 “In that sense, it is similar to the open loop operation system described in [13].” To integrate the comment about the open loop algorithm.

3- The study needs to explain the outdoor daylight and sky conditions in Mercer County, New Jersey, USA that would control the study outcomes.

The authors have added a figure (Figure 3) to explain the outdoor daylight and sky conditions in mercer county throughout the year. Thank you for pointing out this opportunity to drive the point that the reduction of environmental daylight for human comfort is substantial. 

4- The study mentioned about 4 sensors as shown in Figure 1 but I couldn't see the results of each sensor.

The four sensors indicated in Figure 1 are the electric lighting sensors. They are used to adjust the lighting level when the daylight does not allow to reach the 500 lux constraint, see Line 296-298 of initial submission for details. The constraint of the optimization is to produce a 500 lux average work plane and therefore a single system was used to compensate the lack of daylight. The daylight quantity produced by the electric lighting system is given as the difference between the average daylight quantity provided and the objective of 500 lux of work plane illuminance. This value can be assessed by looking at figure 9 that represents the average workplane daylight quantity for the three shading systems and for the three orientations of the test room. 

5- The study mentioned ""The total run-time of the analysis was 73 minutes on an Intel i7 4 core CPU at 4.20GHz. This total time is decomposed in 21 minutes for the daylighting analysis in Radiance, 42 minutes for the thermal analysis in EnergyPlus and 10 minutes for the optimization in MATLAB."

How could this statement help the outcomes of this research?

The authors believe this information provides the reader unfamiliar with one of the aspects of the study with a benchmark of what each section of the methodology requires in terms of computation. In addition, highlighting the ratio between optimization and simulation allows to point out that the methodology is composed of two parts, simulation and optimization. Once the simulation is performed, the optimization can be run independently for varying values of the constraints chosen by the designer. The following sentence was added in section 5.1 to make this point clearer for the reader:

“For a given case study, once the simulations have been performed, the optimization can be run for any value of the illuminance constraint without having to perform the simulations again”

6- Figure 2, model C, it is unclear to understand the model movement and rotation.

The authors have improved the figure to make it more understandable. Namely, the azimuth degree of freedom was clarified and both pictures were made larger.

---

## [Decision Letter · Decision Letter 2]

26 Mar 2020

Occupant-centered optimization framework to evaluate and design new dynamic shading typologies

PONE-D-19-30597R2

Dear Dr. Charpentier,

We are pleased to inform you that your manuscript has been judged scientifically suitable for publication and will be formally accepted for publication once it complies with all outstanding technical requirements.

With kind regards,

Marco Lepidi

Marco Lepidi, PhD

Associate Professor, Academic Editor of PLOS ONE

Dipartimento di Ingegneria Civile, Chimica ed Ambientale

Università degli Studi di Genova (Italy)

Mail: Marco.Lepidi@unige.it

Website: http://www3.dicca.unige.it/mlepidi/index.html

Google Scholar: http://scholar.google.it/citations?user=uePAdVUAAAAJ&hl=it

Additional Editor Comments (optional):

Dear Authors,

two reports have been collected for your re-submission. Both the reports are positive.

Minor improvements are suggested by one of the reviewer.

The manuscript can be accepted for publication.

Best regards

Reviewers' comments:

Reviewer's Responses to Questions

**Comments to the Author**

1. If the authors have adequately addressed your comments raised in a previous round of review and you feel that this manuscript is now acceptable for publication, you may indicate that here to bypass the “Comments to the Author” section, enter your conflict of interest statement in the “Confidential to Editor” section, and submit your "Accept" recommendation.

Reviewer #1: All comments have been addressed

Reviewer #2: All comments have been addressed

2. Is the manuscript technically sound, and do the data support the conclusions?

Reviewer #1: Yes

Reviewer #2: Yes

3. Has the statistical analysis been performed appropriately and rigorously? 

Reviewer #1: N/A

Reviewer #2: Yes

4. Have the authors made all data underlying the findings in their manuscript fully available?

Reviewer #1: Yes

Reviewer #2: Yes

5. Is the manuscript presented in an intelligible fashion and written in standard English?

Reviewer #1: Yes

Reviewer #2: Yes

6. Review Comments to the Author

Reviewer #1: Due to the re-organization of the content of the text, the paper has greatly improved in readability and overall clarity and quality. Authors have addressed all my previous comments. Only a few additional minor changes are suggested:

1. Please only refer to visual comfort (and not lighting comfort as in line 145), specifying that only light quantity is evaluated.

2. Authors are invited to write an additional paragraph summarizing the limitations of the study (before the conclusion section), which are for the moment scattered in the text (e.g., open loop for electric lighting, thermal lag and discrete shading positions, the possibility to only move the slats of the venetian blinds – i.e., they are always covering the window and cannot be completely opened, etc.).

3. Please specify the difference between spherical tracker 1 and 2 in figure 12. I guess the authors use the numbers to indicate the d.o.f. of the system, but the lack of explanation might confuse the reader (it might seem as an additional shading system evaluated). A small drawing indicating which d.o.f. refers to each number might be helpful.

Reviewer #2: The authors have answered all questions and improved the manuscript with clear information and justifications. Therefore no further comments are required.

7. PLOS authors have the option to publish the peer review history of their article (what does this mean?). If published, this will include your full peer review and any attached files.

Reviewer #1: No

Reviewer #2: No

---

## [Editor Report · Acceptance letter]

8 Apr 2020

PONE-D-19-30597R2 

Occupant-centered optimization framework to evaluate and design new dynamic shading typologies 

Dear Dr. Charpentier:

I am pleased to inform you that your manuscript has been deemed suitable for publication in PLOS ONE. Congratulations! Your manuscript is now with our production department. 

With kind regards,

on behalf of

Professor Marco Lepidi 

Academic Editor

PLOS ONE